# WeMix: How to Better Utilize Data Augmentation

## Abstract

Data augmentation is a widely used training trick in deep learning to improve the network generalization ability. Despite many encouraging results, several recent studies did point out limitations of the conventional data augmentation scheme in certain scenarios, calling for a better theoretical understanding of data augmentation. In this work, we develop a comprehensive analysis that reveals pros and cons of data augmentation. The main limitation of data augmentation arises from the data bias, i.e. the augmented data distribution can be quite different from the original one. This data bias leads to a suboptimal performance of existing data augmentation methods. To this end, we develop two novel algorithms, termed "AugDrop" and "MixLoss", to correct the data bias in the data augmentation. Our theoretical analysis shows that both algorithms are guaranteed to improve the effect of data augmentation through the bias correction, which is further validated by our empirical studies. Finally, we propose a generic algorithm "WeMix" by combining AugDrop and MixLoss, whose effectiveness is observed from extensive empirical evaluations.

## 1 Introduction

Data augmentation (Baird, 1992; Schmidhuber, 2015) has been a key to the success of deep learning in image classification (He et al., 2019), and is becoming increasingly common in other tasks such as natural language processing (Zhang et al., 2015) and object detection (Zoph et al., 2019). The data augmentation expands training set by generating virtual instances through random augmentation to the original ones. This alleviates the overfitting (Shorten & Khoshgoftaar, 2019) problem when training large deep neural networks. Despite many encouraging results, it is not the case that data augmentation will always improve generalization errors (Min et al., 2020; Raghunathan et al., 2020). In particular, Raghunathan et al. (2020) showed that training by augmented data will lead to a smaller robust error but potentially a larger standard error. Therefore, it is critical to answer the following two questions before applying data augmentation in deep learning:

- **When will the deep models benefit from data augmentation?**
- **How to better leverage augmented data during training?**

Several previous works (Raghunathan et al., 2020; Wu et al., 2020; Min et al., 2020) tried to address the questions. Their analysis is limited to specific problems such as linear ridge regression therefore may not be applicable to deep learning. In this work, we aim to answer the two questions from a theoretical perspective under a more general non-convex setting. We address the first question in a more general form covering applications in deep learning. For the second question, we develop new approaches that are provably more effective than the conventional data augmentation approaches.

Most data augmentation operations alter the data distribution during the training progress. This imposes a **data distribution bias** (we simply use "data bias" in the rest of this paper) between the augmented data and the original data, which may make it difficult to fully leverage the augmented data. To be more concrete, let us consider label-mixing augmentation (e.g., mixup (Zhang et al., 2018; Tokozume et al., 2018)). Suppose we have $n$ original data $\mathcal{D} = \{(\mathbf{x}_i, \mathbf{y}_i), i = 1, \ldots, n\}$, where the input-label pair $(\mathbf{x}_i, \mathbf{y}_i)$ follows a distribution $\mathbb{P}_{\mathbf{xy}} = (\mathbb{P}_{\mathbf{x}}, \mathbb{P}_{\mathbf{y}}(\cdot|\mathbf{x}))$, $\mathbb{P}_{\mathbf{x}}$ is the marginal distribution of the inputs and $\mathbb{P}_{\mathbf{y}}(\cdot|\mathbf{x})$ is the conditional distribution of the labels given inputs; we generate $m$ augmented data $\widetilde{\mathcal{D}} = \{(\widetilde{\mathbf{x}}_i, \widetilde{\mathbf{y}}_i), i = 1, \ldots, m\}$, where $(\widetilde{\mathbf{x}}_i, \widetilde{\mathbf{y}}_i) \sim \mathbb{P}_{\widetilde{\mathbf{x}}\widetilde{\mathbf{y}}} = (\mathbb{P}_{\widetilde{\mathbf{x}}}, \mathbb{P}_{\widetilde{\mathbf{y}}}(\cdot|\widetilde{\mathbf{x}}))$,

and $\mathbb{P}_{\mathbf{x}} = \mathbb{P}_{\widetilde{\mathbf{x}}}$ but $\mathbb{P}_{\mathbf{y}}(\cdot|\mathbf{x}) \neq \mathbb{P}_{\widetilde{\mathbf{y}}}(\cdot|\widetilde{\mathbf{x}})$. Given $\mathbf{x} \sim \mathbb{P}_{\mathbf{x}}$, the data bias is defined as $\delta_y = \max_{\mathbf{y},\widetilde{\mathbf{y}}} \|\mathbf{y} - \widetilde{\mathbf{y}}\|$. We will show that when the bias between $\mathcal{D}$ and $\widetilde{\mathcal{D}}$ is large, directly training on the augmented data will not be as effective as training on the original data.

Given the fact that augmented data may hurt the performance, the next question is how to design better learning algorithms to leash out the power of augmented data. To this end, we develop two novel algorithms to alleviate the data bias. The first algorithm, termed **AugDrop**, corrects the data bias by introducing a constrained optimization problem. The second algorithm, termed **MixLoss**, corrects the data bias by introducing a modified loss function. We show that, both theoretically and empirically, even with a large data bias, the proposed algorithms can still improve the generalization performance by effectively leveraging the combination of augmented data and original data. We summarize the main contributions of this work as follows:

- We prove that in a conventional training scheme, a deep model can benefit from augmented data when the data bias is small.

- We design two algorithms termed AugDrop and MixLoss that can better leverage augmented data even when the data bias is large with theoretical guarantees.

- Based on our theoretical findings, we empirically propose a new efficient algorithm **WeMix** by combining AugDrop and MixLoss , which has better performances without extra training cost.

## 2 RELATED WORK

A series of empirical works (Cubuk et al., 2019; Ho et al., 2019; Lim et al., 2019; Lin et al., 2019a; Cubuk et al., 2020; Hataya et al., 2019) on how to learn a good policy of using different data augmentations have been proposed without theoretical guarantees. In this section, we mainly focus on reviewing theoretical studies on data augmentation. For a survey of data augmentation, we refer readers to (Shorten & Khoshgoftaar, 2019) and references therein for a comprehensive overview.

Several works have attempted to establish theoretical understandings of data augmentation from different perspectives (Dao et al., 2019; Chen et al., 2019; Rajput et al., 2019). Min et al. (2020) shown that, with more training data, weak augmentation can improve performance while strong augmentation always hurts the performance. Later on, Chen et al. (2020) study the gap between the generalization error (please see the formal definition in (Chen et al., 2020)) of adversarially-trained models and standard models. Both of their theoretical analyses were built on special linear binary classification model or linear regression model for label-preserving augmentation.

Recently, Raghunathan et al. (2020) studied label-preserving transformation in data augmentation, which is identical to the first case in this paper. Their analysis is restricted to linear least square regression under noiseless setting, which is not applicable to training deep neural networks. Besides, their analysis requires infinite unlabeled data. By contrast, we do not require the original data to be unlimited. Wu et al. (2020) considered linear data augmentations. There are several major differences between their work and ours. First, they focus on the ridge linear regression problem which is strongly convex, while we consider non-convex optimization problems, which is more applicable in deep learning. Second, we study more general data augmentations beyond linear transformation.

## 3 PRELIMINARIES AND NOTATIONS

We study a learning problem for finding a classifier to map an input $\mathbf{x} \in \mathcal{X}$ onto a label $\mathbf{y} \in \mathcal{Y} \subset \mathbb{R}^K$, where $K$ is the number of classes. We assume the input-label pair $(\mathbf{x}, \mathbf{y})$ is drawn from a distribution $\mathbb{P}_{\mathbf{xy}} = (\mathbb{P}_{\mathbf{x}}, \mathbb{P}_{\mathbf{y}}(\cdot|\mathbf{x}))$. Since every augmented example $(\widetilde{\mathbf{x}}, \widetilde{\mathbf{y}})$ is generated by applying a certain transformation to either one or multiple examples, we will assume that $(\widetilde{\mathbf{x}}, \widetilde{\mathbf{y}})$ is drawn from a slightly different distribution $\mathbb{P}_{\widetilde{\mathbf{x}}\widetilde{\mathbf{y}}} = (\mathbb{P}_{\widetilde{\mathbf{x}}}, \mathbb{P}_{\widetilde{\mathbf{y}}}(\cdot|\widetilde{\mathbf{x}}))$, where $\mathbb{P}_{\widetilde{\mathbf{x}}}$ is the marginal distribution on the inputs $\widetilde{\mathbf{x}}$ and $\mathbb{P}_{\widetilde{\mathbf{y}}}(\cdot|\widetilde{\mathbf{x}}))$ (we can write it as $\mathbb{P}_{\widetilde{\mathbf{y}}}$ for simplicity) is the conditional distribution of the labels $\widetilde{\mathbf{y}}$ given inputs $\widetilde{\mathbf{x}}$. We sample $n$ training examples $(\mathbf{x}_i, \mathbf{y}_i), i = 1, \ldots, n$ from distribution $\mathbb{P}_{\mathbf{xy}}$ and $m$ training examples $(\widetilde{\mathbf{x}}_i, \widetilde{\mathbf{y}}_i), i = 1, \ldots, m$ from $\mathbb{P}_{\widetilde{\mathbf{x}}\widetilde{\mathbf{y}}}$. We assume that $m \gg n$ due to the data augmentation. We denote by $\mathcal{D} = \{(\mathbf{x}_i, \mathbf{y}_i), i = 1, \ldots, n\}$ and $\widetilde{\mathcal{D}} = (\widetilde{\mathbf{x}}_i, \widetilde{\mathbf{y}}_i), i = 1, \ldots, m\}$ the dataset sampled from $\mathbb{P}_{\mathbf{xy}}$ and $\mathbb{P}_{\widetilde{\mathbf{x}}\widetilde{\mathbf{y}}}$, respectively. We denote by $T(\mathbf{x})$ the set of augmented data transformed from $\mathbf{x}$. We use the notation $\mathrm{E}_{(\mathbf{x},\mathbf{y}) \sim \mathbb{P}_{\mathbf{xy}}}[\cdot]$ to stand for the expectation that takes over a random variable $(\mathbf{x}, \mathbf{y})$ following a distribution $\mathbb{P}_{\mathbf{xy}}$. We denote by $\nabla_{\mathbf{w}} h(\mathbf{w})$ the gradient of a function $h(\mathbf{w})$ in terms of variable $\mathbf{w}$. When the variable to be taken a gradient is obvious, we use

the notation $\nabla h(\mathbf{w})$ for simplicity. Let use $\|\cdot\|$ as the Euclidean norm for a vector or the Spectral norm for a matrix.

The augmented data $\widetilde{\mathcal{D}}$ can be different from the original data $\mathcal{D}$ in two cases, according to (Raghunathan et al., 2020). In the first case, often referred to as **label-preserving**, we consider

$$\mathbb{P}_{\mathbf{y}}(\cdot|\mathbf{x}) = \mathbb{P}_{\widetilde{\mathbf{y}}}(\cdot|\widetilde{\mathbf{x}}), \ \forall \widetilde{\mathbf{x}} \in T(\mathbf{x}) \text{ but } \mathbb{P}_{\mathbf{x}} \neq \mathbb{P}_{\widetilde{\mathbf{x}}}. \tag{1}$$

In the second case, often referred to as **label-mixing**, we consider

$$\mathbb{P}_{\mathbf{x}} = \mathbb{P}_{\widetilde{\mathbf{x}}} \text{ but } \mathbb{P}_{\mathbf{y}}(\cdot|\mathbf{x}) \neq \mathbb{P}_{\widetilde{\mathbf{y}}}(\cdot|\widetilde{\mathbf{x}}), \exists \widetilde{\mathbf{x}} \in T(\mathbf{x}). \tag{2}$$

Examples of label-preserving augmentation include translation, adding noises, small rotation, and brightness or contrast changes (Krizhevsky et al., 2012; Raghunathan et al., 2020). One important example of label-mixing augmentation is mixup (Zhang et al., 2018; Tokozume et al., 2018). Due to the space limitation, we will focus on the label-mixing case, and the related studies and analysis for the label-preserving case can be found in Appendix A. To further quantify the difference between original data and augmented data when $\mathbb{P}_{\mathbf{x}} = \mathbb{P}_{\widetilde{\mathbf{x}}}$ and $\mathbb{P}_{\mathbf{y}} \neq \mathbb{P}_{\widetilde{\mathbf{y}}}$, we introduce the data bias $\delta_y$ given $\mathbf{x} \sim \mathbb{P}_{\mathbf{x}}$ as following:

$$\delta_y := \max_{\mathbf{y}, \widetilde{\mathbf{y}}} \|\mathbf{y} - \widetilde{\mathbf{y}}\|. \tag{3}$$

The equation in (3) measures the difference between the label from original data and the label from augmented data given input $\mathbf{x}$. We aim to learn a prediction function $f(\mathbf{x}; \mathbf{w}) : \mathbb{R}^D \times \mathcal{X} \to \mathbb{R}^K$ that is as close as possible to $\mathbf{y}$, where $\mathbf{w} \in \mathbb{R}^D$ is the parameter and $\mathbb{R}^D$ is a closed convex set. We respectively define two objective functions for optimization problems over the original data and the augmented data as

$$\mathcal{L}(\mathbf{w}) = \mathrm{E}_{(\mathbf{x},\mathbf{y})}\left[\ell\left(\mathbf{y}, f(\mathbf{x}; \mathbf{w})\right)\right], \quad \widetilde{\mathcal{L}}(\mathbf{w}) = \mathrm{E}_{(\widetilde{\mathbf{x}},\widetilde{\mathbf{y}})}\left[\ell\left(\widetilde{\mathbf{y}}, f(\widetilde{\mathbf{x}}; \mathbf{w})\right)\right], \tag{4}$$

where $\ell$ is a cross-entropy loss function which is given by

$$\ell(\mathbf{y}, f(\mathbf{x}; \mathbf{w})) = \sum_{i=1}^{K} y_i p_i(\mathbf{x}; \mathbf{w}), \text{ where } p_i(\mathbf{x}; \mathbf{w}) = -\log\left(\frac{\exp(f_i(\mathbf{x}; \mathbf{w}))}{\sum_{j=1}^{K} \exp(f_j(\mathbf{x}; \mathbf{w}))}\right). \tag{5}$$

We denote by $\mathbf{w}_*$ and $\widetilde{\mathbf{w}}_*$ the optimal solutions to $\min_{\mathbf{w}} \mathcal{L}(\mathbf{w})$ and $\min_{\mathbf{w}} \widetilde{\mathcal{L}}(\mathbf{w})$ respectively,

$$\mathbf{w}_* \in \underset{\mathbf{w} \in \mathbb{R}^D}{\arg\min} \mathcal{L}(\mathbf{w}), \quad \widetilde{\mathbf{w}}_* \in \underset{\mathbf{w} \in \mathbb{R}^D}{\arg\min} \widetilde{\mathcal{L}}(\mathbf{w}). \tag{6}$$

Taking $\mathcal{L}(\mathbf{w})$ as an example, we introduce some function properties used in our analysis.

**Definition 1.** *The stochastic gradients of the objective functions $\mathcal{L}(\mathbf{w})$ is unbiased and bounded, if we have $\mathrm{E}_{(\mathbf{x},\mathbf{y})}\left[\nabla_{\mathbf{w}}\ell\left(\mathbf{y}, f(\mathbf{x}; \mathbf{w})\right)\right] = \nabla\mathcal{L}(\mathbf{w})$, and there exists a constant $G > 0$, such that $\|\nabla_{\mathbf{w}} p(\mathbf{x}; \mathbf{w})\| \leq G, \forall \mathbf{x} \in \mathcal{X}, \forall \mathbf{w} \in \mathbb{R}^D$, where $p(\mathbf{x}; \mathbf{w}) = (p_1(\mathbf{x}; \mathbf{w}), \ldots, p_K(\mathbf{x}; \mathbf{w}))$ is a vector.*

**Definition 2.** *$\mathcal{L}(\mathbf{w})$ is smooth with an $L$-Lipchitz continuous gradient, if there exists a constant $L > 0$ such that $\|\nabla\mathcal{L}(\mathbf{w}) - \nabla\mathcal{L}(\mathbf{u})\| \leq L\|\mathbf{w} - \mathbf{u}\|, \forall \mathbf{w}, \mathbf{u} \in \mathbb{R}^D$, or equivalently, $\mathcal{L}(\mathbf{w}) - \mathcal{L}(\mathbf{u}) \leq \langle\nabla\mathcal{L}(\mathbf{u}), \mathbf{w} - \mathbf{u}\rangle + \frac{L}{2}\|\mathbf{w} - \mathbf{u}\|^2, \forall \mathbf{w}, \mathbf{u} \in \mathbb{R}^D$.*

The above properties are standard and widely used in the literature of non-convex optimization (Ghadimi & Lan, 2013; Yan et al., 2018; Yuan et al., 2019; Wang et al., 2019; Li et al., 2020). We introduce an important property termed Polyak-Łojasiewicz (PL) condition (Polyak, 1963) on the objective function $\mathcal{L}(\mathbf{w})$.

**Definition 3.** *(PL condition) $\mathcal{L}(\mathbf{w})$ satisfies the PL condition, if there exists a constant $\mu > 0$ such that $2\mu(\mathcal{L}(\mathbf{w}) - \mathcal{L}(\mathbf{w}_*)) \leq \|\nabla\mathcal{L}(\mathbf{w})\|^2, \forall \mathbf{w} \in \mathbb{R}^D$, where $\mathbf{w}_*$ is defined in (6).*

The PL condition has been observed in training deep and shallow neural networks (Allen-Zhu et al., 2019; Xie et al., 2017), and is widely used in many non-convex optimization studies (Karimi et al., 2016; Li & Li, 2018; Charles & Papailiopoulos, 2018; Yuan et al., 2019; Li et al., 2020). It is also theoretically verified in (Allen-Zhu et al., 2019) and empirically estimated in (Yuan et al., 2019) for deep neural networks. It is worth noting that PL condition is weaker than many conditions such as strong convexity, restricted strong convexity and weak strong convexity (Karimi et al., 2016).

Finally, we will refer to $\kappa = \frac{L}{\mu}$ as condition number throughout this study.

## 4 MAIN RESULTS

In this section, we present the main results for label-mixing augmentation satisfying (2). Due to the space limitation, we present the results of label-preserving augmentation satisfying (1) in Appendix A. Since we have access to $m \gg n$ augmented data, it is natural to fully leverage the augmented data $\widetilde{\mathcal{D}}$ during training. But on the other hand, due to the data bias $\delta_y$, the prediction model learned from augmented data $\widetilde{\mathcal{D}}$ could be even worse than training the prediction model directly from the original data $\mathcal{D}$, as revealed by Lemma 1 (its proof can be found in Appendix C) and its remark. Throughout this section, suppose that a mini-batch SGD is used for optimization, i.e. to optimize $\mathcal{L}(\mathbf{w})$, we have

$$\mathbf{w}_{t+1} = \mathbf{w}_t - \frac{\eta}{m_0} \sum_{k=1}^{m_0} \nabla_{\mathbf{w}} \ell\left(\mathbf{y}_{k,t}, f(\mathbf{x}_{k,t}; \mathbf{w}_t)\right), \tag{7}$$

where $\eta$ is the step size, $m_0$ is the batch size, and $(\mathbf{x}_{k,t}, \mathbf{y}_{k,t}), k = 1, \ldots, m_0$ are sampled from $\mathcal{D}$. A similar mini-batch SGD algorithm can be developed for the augmented data.

**Lemma 1.** *Assume that $\mathcal{L}$ and $\widetilde{\mathcal{L}}$ satisfy properties in Definition 1, 2 and 3, by setting $\eta = 1/L$ and $m_0 \geq \frac{8}{\delta_y^2}$, when $t \geq \frac{L}{\mu} \log \frac{4(\mathcal{L}(\mathbf{w}_1) - \mathcal{L}(\mathbf{w}_*))\mu}{\delta_y^2 G^2}$, we have*

$$\mathrm{E}[\mathcal{L}(\mathbf{w}_{t+1}) - \mathcal{L}(\mathbf{w}_*)] \leq \delta_y^2 G^2/\mu \leq O(\delta_y^2/\mu), \tag{8}$$

*where $\mathbf{w}_{t+1}$ is output of mini-batch SGD trained on $\widetilde{\mathcal{D}}$, $\delta_y$ is defined in (3).*

**Remark:** It is easy to verify (see the details of proof in Appendix D) that if we simply train the learning model by the original data $\mathcal{D}$, we have

$$\mathrm{E}\left[\mathcal{L}(\mathbf{w}_{n+1}) - \mathcal{L}(\mathbf{w}_*)\right] \leq O\left(L \log(n)/(n\mu^2)\right). \tag{9}$$

Comparing the result in (9) with the result of (8) in Lemma 1, it is easy to show that, when the data bias is too large, i.e., $\delta_y^2 \geq \Omega(L \log(n)/(n\mu))$, we have $O\left(L \log(n)/(n\mu^2)\right) \leq O(\delta_y^2/\mu)$. This implies that training the deep model directly on the original data $\mathcal{D}$ is more effective than on the augmented data $\widetilde{\mathcal{D}}$. Hence, in order to better leverage the augmented data in the presence of large data bias ($\delta_y^2 \geq \Omega(\kappa \log(n)/n)$, where $\kappa = L/\mu$), we need to come up with approaches that automatically correct the data bias. Below, we develop two approaches to correct the data bias. The first approach, termed "AugDrop", corrects the data bias by introducing a constrained optimization approach, and the second approach, termed "MixLoss", addresses the problem by introducing a modified loss function.

### 4.1 **AugDrop**: CORRECTING DATA BIAS BY CONSTRAINED OPTIMIZATION

To address this challenge, we propose a constrained optimization problem, i.e.

$$\min_{w \in \mathbb{R}^D} \mathcal{L}(\mathbf{w}) \quad \text{s.t.} \quad \widetilde{\mathcal{L}}(\mathbf{w}) - \widetilde{\mathcal{L}}(\widetilde{\mathbf{w}}_*) \leq \gamma, \tag{10}$$

where $\gamma > 0$ is a positive constant, $\widetilde{\mathbf{w}}_*$ is defined in (6). The key idea is that by utilizing the augmented data to constrain the solution in a small region, we will be able to enjoy a smaller condition number, leading to a better performance in optimizing $\mathcal{L}(\mathbf{w})$. To make it concrete, we first define three important terms:

$$\gamma_0 := \delta_y^2 G^2/(2\mu), \quad \mathcal{A}(\gamma) = \left\{ \mathbf{w} : \widetilde{\mathcal{L}}(\mathbf{w}) - \widetilde{\mathcal{L}}(\widetilde{\mathbf{w}}_*) \leq \gamma \right\}, \tag{11}$$

$$\mu(\gamma) = \max_{\mu'} \left\{ \mathcal{L}(\mathbf{w}) - \mathcal{L}(\mathbf{w}_*) \leq \|\nabla \mathcal{L}(\mathbf{w})\|^2/(2\mu'), \mathbf{w} \in \mathcal{A}(\gamma) \right\}. \tag{12}$$

We then present a proposition about $\mathcal{A}(\gamma)$ and $\mu(\gamma)$, whose proof is included in Appendix E.

**Proposition 1.** *If $\gamma \in [\gamma_0, 8\gamma_0]$, we have $\mathbf{w}_* \in \mathcal{A}(\gamma)$ and $\mu(\gamma) \geq \mu$.*

According to Proposition 1, by restricting our solutions in $\mathcal{A}(\gamma)$, we have a smaller condition number (since $\mu(\gamma) \geq \mu$) and consequentially a smaller optimization error. It is worth mentioning that the restriction of solutions in $\mathcal{A}(\gamma)$ is reasonable due to the optimal solution $\mathbf{w}_* \in \mathcal{A}(\gamma)$. The idea of using augmentation transformation to restrict the candidate solution was recognized by several

earlier studies, e.g. (Raghunathan et al., 2020). But none of these studies cast it into a constrained optimization problem, a key contribution of our work.

The next question is how to solve the constrained optimization problem in (10). It is worth noting that neither $\mathcal{L}(\mathbf{w})$ nor $\widetilde{\mathcal{L}}(\mathbf{w})$ is convex. Although multiple approaches can be used to solve non-convex constrained optimization problems (Cartis et al., 2011; Lin et al., 2019b; Birgin & Martínez, 2020; Grapiglia & Yuan, 2019; Wright, 2001; O'Neill & Wright, 2020; Boob et al., 2019; Ma et al., 2019), they are too complicated to be implemented in deep learning. Instead, we present a simple approach that divides the optimization into two stages, which is referred to as AugDrop (Please see the details of update steps from Algorithm 2 in Appendix F).

- **Stage I.** We minimize $\widetilde{\mathcal{L}}(\mathbf{w})$ over the augmented data $\widetilde{\mathcal{D}}$. It runs a mini-batch SGD against $\widetilde{\mathcal{D}}$ at least $T_1$ iterations with the size of mini-batch being $m_1$. We denote by $\mathbf{w}_{T_1+1}$ the final output solution of this stage.

- **Stage II.** We minimize $\mathcal{L}(\mathbf{w})$ using the original data $\mathcal{D}$. It initializes the solution $\mathbf{w}_{T_1+1}$ and runs a mini-batch SGD against $\mathcal{D}$ in $n/m_2$ iterations with mini-batch size being $m_2$.

We notice that AugDrop is closely related to TSLA by (Xu et al., 2020) where the first stage trains the data with label smoothing and the second stage trains the data without label smoothing. However, they study the problem how to reduce the variance of stochastic gradient in using label smoothing, while we study how to correct bias in data augmentation by solving a constrained optimization problem. The following theorem states that if we run this two stage optimization algorithm, we could achieve a better performance since $\mu(8\gamma_0)$ is larger than $\mu$. We include its proof in Appendix F.

**Theorem 1.** *Define $\mu_c = \mu(8\gamma_0)$. Assume that $\mathcal{L}$ and $\widetilde{\mathcal{L}}$ satisfy properties in Definition 1, 2 and 3, set learing rate $\eta_1 = 1/L$ in Stage I and learning rate $\eta_2 = \frac{1}{2n\mu_c} \log\left(\frac{8n\mu_c^2(\mathcal{L}(\mathbf{w}_1)-\mathcal{L}(\mathbf{w}_*))}{G^2 L}\right)$ in Stage II for AugDrop. Let $\mathbf{w}_1$ be the initial solution in Stage I of AugDrop and $\mathbf{w}_{T_1+2}, \ldots, \mathbf{w}_{T_1+n/m_2+1}$ be the intermediate solutions obtained by the mini-batch SGD in Stage II of AugDrop. Choose $T_1 = \frac{1}{\eta_1\mu} \log \frac{2(\widetilde{\mathcal{L}}(\mathbf{w}_1)-\widetilde{\mathcal{L}}(\widetilde{\mathbf{w}}_*))\mu}{\delta_y^2 G^2}$, $m_1 = \left(1 + \sqrt{3 \log \frac{2T_1}{\delta}}\right)^2 \frac{8}{\delta_y^2}$ and $m_2 = \left(1 + \sqrt{3 \log \frac{2n}{\delta}}\right)^2 \frac{4}{\delta_y^2}$, with a probability $1 - \delta$, we have $\mathbf{w}_t \in \mathcal{A}(8\gamma_0), \forall t \in \{T_1 + 2, \ldots, T_1 + n/m_2 + 1\}$ and*

$$\mathrm{E}\left[\mathcal{L}(\widehat{\mathbf{w}}) - \mathcal{L}(\mathbf{w}_*)\right] \leq \frac{G^2 L}{4n\mu_c^2}\left(1 + \log\left(\frac{4n\mu_c^2(\mathcal{L}(\mathbf{w}_1)-\mathcal{L}(\mathbf{w}_*))}{G^2 L}\right)\right) \leq O\left(\frac{L\log(n)}{n\mu_c^2}\right), \quad (13)$$

*where $\widehat{\mathbf{w}} = \mathbf{w}_{T_1+n/m_2+1}$ and $\delta_y$ is defined in in (3).*

**Remark.** Theorem 1 shows that all intermediate solutions $\mathbf{w}_t$ obtained in Stage II of AugDrop satisfy the constraint $\widetilde{\mathcal{L}}(\mathbf{w}_t) - \widetilde{\mathcal{L}}(\widetilde{\mathbf{w}}_*) \leq 8\gamma_0$, that is to say, $\mathbf{w}_t \in \mathcal{A}(8\gamma_0)$. Based on Proposition 1, we will enjoy a larger $\mu_c$ than $\mu$. Comparing the result of (13) in Theorem 1 with (9), training by using AugDrop will result in a better performance than directly training on $\mathcal{D}$ due to $\mu_c \geq \mu$. Besides, when the data bias is large, i.e., $\delta_y^2 \geq \Omega(L\log(n)/(n\mu))$, we know $O(L\log(n)/(n\mu_c^2)) \leq O(\mu\delta_y^2/\mu_c^2) \leq O(\delta_y^2/\mu)$, where the last inequality holds due to $\mu_c \geq \mu$. By comparing (13) with the result of (8) in Lemma 1, we know that training by using AugDrop has a better performance than directly training on $\widetilde{\mathcal{D}}$ when the data bias is large. By solving a constrained problem, the AugDrop algorithm can correct the data bias and thus can enjoy an better performance.

## 4.2 **MixLoss**: Correcting Data Bias by Modified Loss Function

Without loss of generality, we set $\mathcal{L}(\mathbf{w}_*) = 0$, a common property observed in training deep neural networks (Zhang et al., 2016; Allen-Zhu et al., 2019; Du et al., 2018; 2019; Arora et al., 2019; Chizat et al., 2019; Hastie et al., 2019; Yun et al., 2019; Zou et al., 2020). Since $\|\mathbf{y} - \widetilde{\mathbf{y}}\| \leq \delta_y$ for any $\mathbf{y}$ and $\widetilde{\mathbf{y}}$ and given $\mathbf{x}$, we define a new loss function $\ell_a(\widetilde{\mathbf{y}}, f(\widetilde{\mathbf{x}}; \mathbf{w}))$ as

$$\ell_a(\widetilde{\mathbf{y}}, f(\widetilde{\mathbf{x}}; \mathbf{w})) = \min_{\|\mathbf{z}-\widetilde{\mathbf{y}}\| \leq \delta_y} \ell(\mathbf{z}, f(\widetilde{\mathbf{x}}; \mathbf{w})). \quad (14)$$

It has been shown that since the cross-entropy loss $\ell(\mathbf{z}, \cdot)$ is convex in terms of $\mathbf{z} \in \mathcal{Y}$, then the minimization problem (14) is a convex optimization problem and has a closed form solution (Boyd & Vandenberghe, 2004). Using this new loss, we define a new objective function $\mathcal{L}_a(\mathbf{w})$

$$\mathcal{L}_a(\mathbf{w}) = \mathrm{E}_{(\widetilde{\mathbf{x}},\widetilde{\mathbf{y}})}\left[\ell_a(\widetilde{\mathbf{y}}, f(\widetilde{\mathbf{x}}; \mathbf{w}))\right] = \mathrm{E}_{(\widetilde{\mathbf{x}},\widetilde{\mathbf{y}})}\left[\min_{\|\mathbf{z}-\widetilde{\mathbf{y}}\| \leq \delta_y} \ell(\mathbf{z}, f(\widetilde{\mathbf{x}}; \mathbf{w}))\right]. \quad (15)$$

---

**Algorithm 1** WeMix

---

1: **Input:** $T_1, T_2$, stochastic algorithms $\mathcal{A}_1, \mathcal{A}_2$ (e.g., momentum SGD, SGD)
2: **Initialize**: $\mathbf{w}_1 \in \mathbb{R}^D$, $\lambda \in (0, 1)$, $\eta_1, \eta_2 > 0$
 // First stage: Weighted Mixed Losses
3: **for** $t = 1, 2, \ldots, T_1$ **do**
4:    draw examples $(\mathbf{x}_{i_t}, \mathbf{y}_{i_t})$ at random from training data   $\diamond$ construct stochastic gradient of $\mathcal{L}$
5:    generate augmented examples $(\widetilde{\mathbf{x}}_{j_t}, \widetilde{\mathbf{y}}_{j_t})$        $\diamond$ construct stochastic gradient of $\mathcal{L}_a$
6:    compute stochastic gradient $\widehat{\mathbf{g}}_t = \lambda \nabla \ell(\mathbf{y}_{i_t}, f(\mathbf{x}_{i_t}; \mathbf{w}_t)) + (1 - \lambda) \nabla \ell_a(\widetilde{\mathbf{y}}_{i_t}, f(\widetilde{\mathbf{x}}_{i_t}; \mathbf{w}_t))$
7:    $\mathbf{w}_{t+1} = \mathcal{A}_1(\mathbf{w}_t; \widehat{\mathbf{g}}_t, \eta_1)$                     $\diamond$ update one step of $\mathcal{A}_1$
8: **end for**
 // Second stage: Augmentation Dropping
9: **for** $t = T_1 + 1, T_1 + 2, \ldots, T_1 + T_2$ **do**
10:    draw examples $(\mathbf{x}_{i_t}, \mathbf{y}_{i_t})$ at random from training data   $\diamond$ construct stochastic gradient of $\mathcal{L}$
11:    compute stochastic gradient $\widehat{\mathbf{g}}_t = \nabla \ell(\mathbf{y}_{i_t}, f(\mathbf{x}_{i_t}; \mathbf{w}_t))$
12:    $\mathbf{w}_{t+1} = \mathcal{A}_2(\mathbf{w}_t; \widehat{\mathbf{g}}_t, \eta_2)$                     $\diamond$ update one step of $\mathcal{A}_2$
13: **end for**
14: **Output:** $w_{T_1+T_2+1}$.

---

It is easy to verify that $\mathcal{L}_a(\mathbf{w}_*) = 0$ and therefore $\mathbf{w}_*$ also minimizes $\mathcal{L}_a(\mathbf{w})$ (see Appendix G). In contrast, $\widetilde{\mathbf{w}}_*$, the minimizer of $\widetilde{\mathcal{L}}(\mathbf{w})$, can be very different from $\mathbf{w}_*$. Hence, we can correct the data bias arising from the augmented data by replacing $\widetilde{\mathcal{L}}(\mathbf{w})$ with $\mathcal{L}_a(\mathbf{w})$, leading to the following optimization problem:

$$\min_{\mathbf{w} \in \mathbb{R}^D} \mathcal{L}_c(\mathbf{w}) = \lambda \mathcal{L}(\mathbf{w}) + (1 - \lambda) \mathcal{L}_a(\mathbf{w}), \tag{16}$$

where $\lambda \in (0, 1)$. Since $\mathcal{L}_c(\mathbf{w})$ shares the same minimizer with $\mathcal{L}(\mathbf{w})$ (see Appendix G), it is sufficient to optimize $\mathcal{L}_c(\mathbf{w})$, instead of optimizing $\mathcal{L}(\mathbf{w})$. The main advantage of minimizing $\mathcal{L}_c(\mathbf{w})$ over $\mathcal{L}(\mathbf{w})$ is that by introducing a small $\lambda$, we will be able to reduce the variance in computing the gradient of $\mathcal{L}_c(\mathbf{w})$, and therefore improve the overall convergence. More specifically, our SGD method is given as follows: at each iteration $t$, we compute the approximate gradient as $\widehat{\mathbf{g}}_t = \lambda \nabla \ell(\mathbf{y}_t, f(\mathbf{x}_t; \mathbf{w}_t)) + (1 - \lambda) \frac{1}{m_0} \sum_{i=1}^{m_0} \nabla \ell_a(\widetilde{\mathbf{y}}_{t,i}, f(\widetilde{\mathbf{x}}_{t,i}; \mathbf{w}_t))$, where $(\mathbf{x}_t, \mathbf{y}_t)$ is an example sampled from $\mathcal{D}$ at iteration $t$. We refer to this approach as MixLoss (Please see the details of update steps from Algorithm 3 in Appendix H). We then give the convergence result in the following theorem, whose proof is included in Appendix H.

**Theorem 2.** *Assume that $\mathcal{L}$, $\widetilde{\mathcal{L}}$ and $\mathcal{L}_a$ satisfy properties in Definition 1, 2 and 3, by setting $m_0 \geq \frac{72(1-\lambda)^2}{\lambda^2}$ and $\eta = \frac{1}{\mu n} \log \frac{n \mu^2 \mathcal{L}(\mathbf{w}_1)}{\lambda^2 L G^2} \leq \frac{1}{2L}$ in MixLoss, we have*

$$\mathrm{E}\left[\mathcal{L}(\mathbf{w}_{n+1}) - \mathcal{L}(\mathbf{w}_*)\right] \leq \frac{\lambda L G^2}{n \mu^2} \left(1 + 5 \log \frac{n \mu^2 \mathcal{L}(\mathbf{w}_1)}{\lambda^2 L G^2}\right) \leq O\left(\frac{\lambda L \log(n/\lambda^2)}{n \mu^2}\right). \tag{17}$$

**Remark.** According to the results in (17) and (9), we know that $O\left(\lambda L \log(n/\lambda^2)/(n\mu^2)\right) \leq O\left(L \log(n)/(n\mu^2)\right)$ when an appropriate $\lambda \in (0, 1)$ is selected, leading to a better performance by using MixLoss compared with the performance trained on the original data $\mathcal{D}$. For example, one can simply use $\lambda = O(\mu/L)$. On the other hand, when the data bias is large where $\delta_y^2$ satisfying $\delta_y^2 \geq \Omega(L \log(n)/(n\mu))$, we know $O(L \log(n)/(n\mu^2)) \leq O(\delta_y^2/\mu)$. Based on previous discussion, by choosing an appropriate $\lambda \in (0, 1)$ (e.g., $\lambda = O(\mu/L)$), we will have $O\left(\lambda L \log(n/\lambda^2)/(n\mu^2)\right) \leq O(\delta_y^2/\mu)$. Then by comparing (17) with (8), we know that training by using MixLoss has a better performance than directly training on $\widetilde{\mathcal{D}}$ when the data bias is large. Therefore, by solving the problem with a modified loss function, the MixLoss algorithm can enjoy a better performance by correcting the data bias.

### 4.3 WeMix: A Generic Weighted Mixed Losses with Augmentation Dropping Algorithm

Inspired by previous theoretical analysis of using augmented data, we propose a generic framework of weighted mixed losses with augmentation dropping that builds upon two algorithms, Aug-Drop and MixLoss. Algorithm 1 describes our procedure in detail, which is referred to as WeMix. It consists of two stages, wherein the first stage it runs a stochastic algorithm $\mathcal{A}_1$ (e.g., momentum SGD, SGD) for solving weighted mixed losses (16) and the second stage it runs another/same

Table 1: Comparison of Testing Top-1 Accuracy (mean $\pm$ standard deviation, in %) using Different Methods on ResNet-18 over CIFAR-10 and CIFAR-100 for mixup

| Method | CIFAR-100 | CIFAR-10 |
|---|---|---|
| without mixup | $76.97 \pm 0.27$ | $94.95 \pm 0.17$ |
| mixup | $78.31 \pm 0.18$ | $95.67 \pm 0.09$ |
| AugDrop (ours) | $80.24 \pm 0.34$ | $96.03 \pm 0.12$ |
| MixLoss (ours) | $79.70 \pm 0.31$ | $95.94 \pm 0.11$ |
| WeMix (ours) | $\mathbf{80.61} \pm 0.10$ | $\mathbf{96.11} \pm 0.11$ |
| MixLoss-s (ours) | $79.53 \pm 0.13$ | $95.87 \pm 0.14$ |
| WeMix-s (ours) | $80.29 \pm 0.22$ | $96.06 \pm 0.16$ |

stochastic algorithm $\mathcal{A}_2$ (e.g., momentum SGD, SGD) for solving the problem over original data. The notation $\mathcal{A}(\cdot; \cdot, \eta)$ is one update step of a stochastic algorithm $\mathcal{A}$ with learning rate $\eta$. For example, if we select SGD as algorithm $\mathcal{A}$, then $\text{SGD}(\mathbf{w}_t; \widehat{\mathbf{g}}_t, \eta) = \mathbf{w}_t - \eta\widehat{\mathbf{g}}_t$. The proposed WeMix is a generic strategy where the subroutine algorithm $\mathcal{A}_1/\mathcal{A}_2$ can be replaced by any stochastic algorithms such as stochastic versions of momentum methods (Polyak, 1964; Nesterov, 1983; Yan et al., 2018) and adaptive methods (Duchi et al., 2011; Hinton et al., 2012; Zeiler, 2012; Kingma & Ba, 2015; Dozat, 2016; Reddi et al., 2018). We can also replace $\ell_a$ by $\ell$ to avoid solving a minimization problem. The last solution of the first stage will be used as the initial solution of the second stage. If $\lambda = 0$ and $\ell_a = \ell$, then WeMix reduces to the AugDrop; while if $T_2 = 0$, WeMix becomes to MixLoss. For label-preserving case, we only need to simply use $\ell_a = \ell$ (i.e, $\delta_f = 0$) in WeMix.

## 5 EXPERIMENTS

To evaluate the performance of the proposed methods, we trained deep neural networks on two benchmark data sets, CIFAR-10 and CIFAR-100[1] (Krizhevsky & Hinton, 2009) for the image classification task. Both CIFAR-10 and CIFAR-100 have 50,000 training images and 10,000 testing images of $32 \times 32$ resolutions. CIFAR-10 has 10 classes containing 6000 images each, while CIFAR-100 has 100 classes. We use mixup (Zhang et al., 2018) as an example of lable-mixing augmentation and Contrast as an example of lable-preserving augmentation and. For the choice of backbone, we use ResNet-18 model (He et al., 2016) in mixup, and Wide-ResNet-28-10 model (Zagoruyko & Komodakis, 2016) is applied in the Contrast experiment following by (Cubuk et al., 2019; 2020). To verify our theoretical results, we compare the proposed AugDrop and MixLoss with two baselines, SGD with mixup/Contrast and SGD without mixup/Contrast (baseline). We also include WeMix in the comparison. The mini-batch size of training instances for all methods is 256 as suggested by He et al. (2019) and He et al. (2016). The momentum parameter of 0.9 is used. The weight decay with the parameter value is set to be $5 \times 10^{-4}$. The total epochs of training progress is fixed as 200. Followed by (He et al., 2016; Zagoruyko & Komodakis, 2016), we use 0.1 as the initial learning rates for all algorithms and divide them by 10 every 60 epochs.

For AugDrop, we drop off the augmentation after $s$-th epoch, where $s \in \{150, 160, 170, 180, 190\}$ is tuned. For example, if $s = 160$, then it means that we run the first stage of AugDrop 160 epochs and the second stage 40 epochs. For MixLoss, we tune the parameter $\delta_y$ from $\{0.5, 0.05, 0.005, 0.0005\}$ and the best performance is reported. For WeMix, we use the value of $\delta_y$ with the best performance in MixLoss, and we tune the dropping off epochs $s$ same as AugDrop. We fix the convex combination parameter $\lambda = 0.1$ both for MixLoss and WeMix. We use top-1 accuracy to evaluate the performance. All top-1 accuracy on the testing data set are averaged over 5 independent random trails with their standard deviations.

### 5.1 MIXUP

Given two examples $(\mathbf{x}_i, \mathbf{y}_i)$ and $(\mathbf{x}_j, \mathbf{y}_j)$ that are drawn at random from the training data, mixup creates a virtual training example as follows $\mathbf{x}' = \beta\mathbf{x}_i + (1 - \beta)\mathbf{x}_j, \mathbf{y}' = \beta\mathbf{y}_i + (1 - \beta)\mathbf{y}_j$, where $\beta \in [0, 1]$ is sampled from a Beta distribution $\beta(\alpha, \alpha)$. We use $\alpha = 1$ in the experiments as suggested in (Zhang et al., 2018). In this subsection, we want to empirically verify that our theoretical findings for label-mixing augmentation in Section 4. The experimental results conducted on CIFAR-10 and CIFAR-100 are listed in Table 1. We can see from the results that both AugDrop and MixLoss are

---

[1] https://www.cs.toronto.edu/~kriz/cifar.html

Table 2: Comparison of Testing Top-1 Accuracy (mean $\pm$ standard deviation, in %) using Different Methods on ResNet-18 over CIFAR-100 for mixup of three images and ten images

| Method | 3 images | 10 images |
|--------|----------|-----------|
| Mixup | $76.56 \pm 0.23$ | $60.36 \pm 0.88$ |
| AugDrop | $80.18 \pm 0.19$ | $76.35 \pm 0.27$ |
| MixLoss | $79.61 \pm 0.09$ | $75.41 \pm 0.19$ |
| WeMix | $80.41 \pm 0.22$ | $78.08 \pm 0.11$ |

Table 3: Comparison of Testing Top-1 Accuracy (mean $\pm$ standard deviation, in %) using Different Methods on WideResNet-28-10 over CIFAR-10 and CIFAR-100 for Contrast Transformation

| Method | CIFAR-100 | CIFAR-10 |
|--------|-----------|----------|
| without Contrast | $78.07 \pm 0.27$ | $95.51 \pm 0.14$ |
| Contrast | $77.90 \pm 0.26$ | $95.66 \pm 0.05$ |
| AugDrop (ours) | $78.40 \pm 0.24$ | $\mathbf{95.93} \pm 0.21$ |
| MixLoss (ours) | $78.17 \pm 0.20$ | $95.70 \pm 0.11$ |
| WeMix (ours) | $\mathbf{78.79} \pm 0.18$ | $95.81 \pm 0.11$ |

better than two baselines, with and without mixup, which matches the theory found in Section 4. The performance of MixLoss is slightly worse than that of AugDrop, but they are comparable. Besides, the proposed WeMix enjoys both improvements, leading to the best performance among all algorithms although its convergence theoretical guarantee is unclear.

Next, we implement MixLoss and WeMix with $\delta_y = 0$ (i.e., use $\ell_a = \ell$), which are denoted by MixLoss-s and WeMix-s, respectively. We summarize the results in Table 1, showing that both MixLoss-s and WeMix-s drop performance, comparing with MixLoss and WeMix, respectively.

Besides, we use more than two images in mixup such as three and ten images and the results are shown in Table 2. Although the top-1 accuracy of mixup reduces dramatically, we find that the proposed WeMix can still improve the performance when it comparing with mixup itself, showing the robustness of WeMix.

## 5.2 Contrast

As a simple label-preserving augmentation, Contrast controls the contrast of the image. Its transformation magnitude is randomly selected from a uniform distribution $[0.1, 1.9]$ following by (Cubuk et al., 2019). Despite its simplicity, we choose it to demonstrate our theory for the considered case in Appendix A. The results of highest top-1 accuracy on the testing data sets for different methods are presented in Table 3. We find that by directly training on data with Contrast, it will drop the performance a little bit. Even so, the result shows that AugDrop has better performance than two baselines, which is consistent with the theoretical findings for label-preserving augmentation in Appendix A that we need use the data augmentation at the early training stage but drop it at the end of training. Although there is no theoretical guarantee for the label-preserving transformation case, we implement MixLoss and WeMix by setting $\delta_f = 0$, i.e., using $\ell_a = \ell$ in (14). The results show that MixLoss and WeMix are better than two baselines but are slightly worse than AugDrop.

## 6 Conclusions and Future Work

In this paper, we have studied how to better utilize data augmentation in training deep neural networks by designing two training schemes with the first one switches augmented data to original data during the training progress and the second one training on a convex combination of original loss and augmented loss. We have provided theoretical analyses of these two training schemes in non-convex smooth optimization setting. With the insights of theoretical results, we have designed a generic algorithm WeMix that can well leverage data augmentation in practice. We have verified our theoretical finding throughout extensive experimental evaluations on training ResNet and WideResNet models over benchmark data sets. Despite the effectiveness of WeMix, its theoretical guarantee is still not fully understand. We would like to leave this open problem as future work.

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

# A    MAIN RESULTS FOR LABEL-PRESERVING AUGMENTATION

We consider label-preserving augmentation case (1), that is,

$$\mathbb{P}_{\mathbf{y}}(\cdot|\mathbf{x}) = \mathbb{P}_{\widetilde{\mathbf{y}}}(\cdot|\widetilde{\mathbf{x}}), \ \forall \widetilde{\mathbf{x}} \in T(\mathbf{x}) \text{ but } \mathbb{P}_{\mathbf{x}} \neq \mathbb{P}_{\widetilde{\mathbf{x}}}.$$

It covers many image data augmentations including translation, adding noises, small rotation, and brightness or contrast changes (Krizhevsky et al., 2012; Raghunathan et al., 2020). It is worth mentioning that the compositions of label-preserving augmentation could also be label-preserving. Similar to the case of label-mixing augmentation, we measure the following difference between $\mathbb{P}_{\mathbf{x}}$ and $\mathbb{P}_{\widetilde{\mathbf{x}}}$ by a KL divergence:

$$\delta_P := D_{KL}(\mathbb{P}_{\mathbf{x}} \| \mathbb{P}_{\widetilde{\mathbf{x}}}) = \mathrm{E}_{\mathbf{x} \sim \mathbb{P}_x} \left[ \log \frac{\mathbb{P}_{\mathbf{x}}(\mathbf{x})}{\mathbb{P}_{\widetilde{\mathbf{x}}}(\mathbf{x})} \right]. \tag{18}$$

Due to the **data bias** $\delta_P$, the prediction model learned from augmented data $\widetilde{\mathcal{D}}$ could be even worse than training the prediction model directly from the original data $\mathcal{D}$, as revealed by the following lemma and its remark.

**Lemma 2.** *(label-preserving augmentation) Assume that $\mathcal{L}$ and $\widetilde{\mathcal{L}}$ satisfy properties in Definition 1, 2 and 3, by setting $\eta = 1/L$ and $m_0 \geq \frac{4}{\eta \delta_P}$, when $t \geq t_0 = \frac{L}{\mu} \log \frac{(\mathcal{L}(\mathbf{w}_1) - \mathcal{L}(\mathbf{w}_*))\mu}{2\delta_P G^2}$, we have*

$$\mathrm{E}_{(\widetilde{\mathbf{x}}_t, \widetilde{\mathbf{y}}_t)}[\mathcal{L}(\mathbf{w}_{t+1}) - \mathcal{L}(\mathbf{w}_*)] \leq \frac{4\delta_P G^2}{\mu} \leq O\left(\frac{\delta_P}{\mu}\right), \tag{19}$$

*where $\mathbf{w}_{t+1}$ is output of mini-batch SGD trained on $\widetilde{\mathcal{D}}$, $\delta_P$ is defined in (18).*

*Proof.* See Appendix I.1.                                                                        □

**Remark:** Comparing the result in (9) with the result of (19) in Lemma 2, it is easy to show that, when the data bias is too large, i.e., $\delta_P \geq \Omega(L \log(n)/(n\mu))$, we have $O\left(L \log(n)/(n\mu^2)\right) \leq O(\delta_P/\mu)$. This implies that training the deep model directly on the original data $\mathcal{D}$ is more effective than on the augmented data $\widetilde{\mathcal{D}}$. Hence, in order to better leverage the augmented data in the presence of large data bias ($\delta_P \geq \Omega(\kappa \log(n)/n)$, where $\kappa = L/\mu$), we need to come up with an approach that automatically correct the data bias in $\widetilde{\mathcal{D}}$. Below, we use AugDrop to correct the data bias by solving a constrained optimization problem.

## A.1    **AugDrop**: CORRECTING DATA BIAS BY CONSTRAINED OPTIMIZATION

To correct data bias, we consider to solve the constrained optimization problem (10). The key idea is to shrink the solution in a small region by using utilize augmented data to enjoy a smaller condition number, leading to an improved convergence in optimizing $\mathcal{L}(\mathbf{w})$. By introducing a term that

$$\gamma_1 := \delta_P G^2/\mu,$$

we can present a proposition about $\mathcal{A}(\gamma)$ and $\mu(\gamma)$, showing that we have a smaller condition number and consequentially a smaller optimization error by restricting our solutions to $\mathcal{A}(\gamma)$.

**Proposition 2.** *If $\gamma \in [\gamma_1, 4\gamma_1]$, we have $w_* \in \mathcal{A}(\gamma)$ and $\mu(\gamma) \geq \mu$, where $\mathcal{A}(\gamma)$ and $\mu(\gamma)$ are defined in (11) and (12), respectively.*

*Proof.* See Appendix I.2.                                                                        □

The following theorem shows the convergence result of AugDrop for label-preserving augmentation.

**Theorem 3.** *Define $\gamma = 4\gamma_1, \mu_e = \mu(4\gamma_1)$. Assume that $\mathcal{L}$ and $\widetilde{\mathcal{L}}$ satisfy properties in Definition 1, 2 and 3, set learning rate $\eta_1 = 1/L$ in Stage I and learning rate $\eta_2 = \frac{1}{2n\mu_e} \log \left( \frac{8n\mu_e^2(\mathcal{L}(\mathbf{w}_1) - \mathcal{L}(\mathbf{w}_*))}{G^2 L} \right)$ in Stage II for AugDrop. Let $\mathbf{w}_1$ be the initial solution in Stage I of AugDrop and $\mathbf{w}_{T_1+2}, \ldots, \mathbf{w}_{T_1+n/m_2+1}$ be the intermediate solutions obtained by the mini-batch SGD in Stage II of AugDrop. Choose $T_1 = \frac{L}{\mu} \log \frac{2(\widetilde{\mathcal{L}}(\mathbf{w}_1) - \widetilde{\mathcal{L}}(\widetilde{\mathbf{w}}_*))\mu}{\delta_P G^2}, m_1 = \left( 1 + \sqrt{3 \log \frac{2T_1}{\delta}} \right)^2 \frac{8}{\delta_P}$*

and $m_2 = \left(1 + \sqrt{3 \log \frac{2n}{\delta}}\right)^2 \frac{8}{\delta_P}$, with a probability $1 - \delta$, we have $\mathbf{w}_t \in \mathcal{A}(4\gamma_1), \forall t \in \{T_1 + 2, \ldots, T_1 + n/m_2 + 1\}$ and

$$\mathrm{E}\left[\mathcal{L}(\widehat{\mathbf{w}}) - \mathcal{L}(\mathbf{w}_*)\right] \leq \frac{G^2 L}{8n\mu_e^2} + \frac{G^2 L}{8n\mu_e^2} \log\left(\frac{8n\mu_e^2(\mathcal{L}(\mathbf{w}_1) - \mathcal{L}(\mathbf{w}_*))}{G^2 L}\right) \leq O\left(\frac{L \log(n)}{n\mu_e^2}\right), \quad (20)$$

where $\widehat{\mathbf{w}} = \mathbf{w}_{T_1 + n/m_2 + 1}$ and $\delta_P$ is defined in (18).

*Proof.* See Appendix I.3. $\qquad\qquad\square$

**Remark.** Theorem 3 shows that all intermediate solutions $\mathbf{w}_t$ obtained in Stage II of AugDrop satisfy the constraint $\widetilde{\mathcal{L}}(\mathbf{w}_t) - \widetilde{\mathcal{L}}(\widetilde{\mathbf{w}}_*) \leq 4\gamma_1$, that is to say, $\mathbf{w}_t \in \mathcal{A}(4\gamma_1)$. Based on Proposition 2, we will enjoy a larger $\mu_e$ than $\mu$. Comparing the result of (20) in Theorem 3 with (9), training by using AugDrop will result in a better performance than directly training on $\mathcal{D}$ due to $\mu_e \geq \mu$. Besides, when the data bias is large, i.e., $\delta_P \geq \Omega(L \log(n)/(n\mu))$, we know $O(L \log(n)/(n\mu_e^2)) \leq O(\mu\delta_P/\mu_e^2) \leq O(\delta_P/\mu)$, where the last inequality holds due to $\mu_e \geq \mu$. By comparing (20) with the result of (19) in Lemma 2, we know that training by using AugDrop has a better performance than directly training on $\widetilde{\mathcal{D}}$ when the data bias is large. By solving a constrained problem, the AugDrop algorithm can correct the data bias and thus enjoy an better performance.

## B   Technical Results for Cross-entropy Loss

**Lemma 3.** *Assume that $\mathcal{L}(\mathbf{w}) = \mathrm{E}[\ell(\mathbf{y}, f(\mathbf{x}; \mathbf{w}))]$ satisfies property in Definition 1, where $\ell$ is a cross-entropy loss, then we have*

$$\|\nabla_{\mathbf{w}}\ell(\mathbf{y}, f(\mathbf{x}; \mathbf{w})) - \nabla_{\mathbf{w}}\ell(\widetilde{\mathbf{y}}, f(\mathbf{x}; \mathbf{w}))\| \leq G\|\mathbf{y} - \widetilde{\mathbf{y}}\|, \quad (21)$$

*and*

$$\|\nabla_{\mathbf{w}}\ell(\mathbf{y}, f(\mathbf{x}; \mathbf{w}))\| \leq G. \quad (22)$$

*Proof.* The objective function is

$$\mathcal{L}(\mathbf{w}) = \mathrm{E}_{(\mathbf{x}, \mathbf{y})}\left[\ell(\mathbf{y}, f(\mathbf{x}; \mathbf{w}))\right], \quad (23)$$

where the cross-entropy loss function $\ell$ is given by

$$\ell(\mathbf{y}, f(\mathbf{x}; \mathbf{w})) = \sum_{i=1}^{K} -y_i \log\left(\frac{\exp(f_i(\mathbf{x}; \mathbf{w}))}{\sum_{j=1}^{K} \exp(f_j(\mathbf{x}; \mathbf{w}))}\right). \quad (24)$$

Let set

$$p(\mathbf{x}; \mathbf{w}) = (p_1(\mathbf{x}; \mathbf{w}), \ldots, p_K(\mathbf{x}; \mathbf{w})), \quad p_i(\mathbf{x}; \mathbf{w}) = -\log\left(\frac{\exp(f_i(\mathbf{x}; \mathbf{w}))}{\sum_{j=1}^{K} \exp(f_j(\mathbf{x}; \mathbf{w}))}\right), \quad (25)$$

then the gradient of $\ell$ with respective to $w$ is

$$\nabla\ell(\mathbf{y}, f(\mathbf{x}; \mathbf{w})) = \langle \mathbf{y}, \nabla p(\mathbf{x}; \mathbf{w}) \rangle. \quad (26)$$

Therefore, $\forall \mathbf{x} \in \mathcal{X}$ and $\mathbf{w} \in \mathbb{R}^D$ we have

$$\begin{aligned}
&\|\nabla_{\mathbf{w}}\ell(\mathbf{y}, f(\mathbf{x}; \mathbf{w})) - \nabla_{\mathbf{w}}\ell(\widetilde{\mathbf{y}}, f(\mathbf{x}; \mathbf{w}))\| \\
&= \|\langle \mathbf{y} - \widetilde{\mathbf{y}}, \nabla p(\mathbf{x}; \mathbf{w}) \rangle\| \\
&\leq \|\nabla p(\mathbf{x}; \mathbf{w})\| \|\mathbf{y} - \widetilde{\mathbf{y}}\| \\
&\leq G\|\mathbf{y} - \widetilde{\mathbf{y}}\|,
\end{aligned} \quad (27)$$

and

$$\|\nabla_{\mathbf{w}}\ell(\mathbf{y}, f(\mathbf{x}; \mathbf{w}))\| = \|\langle \mathbf{y}, \nabla p(\mathbf{x}; \mathbf{w}) \rangle\| \leq \|\nabla p(\mathbf{x}; \mathbf{w})\| \|\mathbf{y}\| \leq G, \quad (28)$$

where uses the facts that $\|\nabla p(\mathbf{x}; \mathbf{w})\| \leq G$ and $\|\mathbf{y}\| \leq \|\mathbf{y}\|_1 = 1$, here $\|\cdot\|$ is a Euclidean norm ($\ell_2$ norm) and $\|\cdot\|_1$ is $\ell_1$ norm. $\qquad\square$

## C    PROOF OF LEMMA 1

*Proof.* Recall that the update of mini-batch SGD is given by

$$\mathbf{w}_{t+1} = \mathbf{w}_t - \eta \widetilde{\mathbf{g}}_t.$$

Let set the averaged mini-batch stochastic gradients of $\widetilde{\mathcal{L}}(\mathbf{w}_t)$ as

$$\widetilde{\mathbf{g}}_t := \frac{1}{m_0} \sum_{i=1}^{m_0} \nabla \ell \left( \widetilde{\mathbf{y}}_{t,i}, f(\widetilde{\mathbf{x}}_{t,i}; \mathbf{w}_t) \right),$$

then by the Assumption of $\widetilde{\mathcal{L}}$ satisfying the property in Definition 1, we know that

$$\mathrm{E}_{(\widetilde{\mathbf{x}}_{t,i}, \widetilde{\mathbf{y}}_{t,i})} \left[ \nabla \ell \left( \widetilde{\mathbf{y}}_{t,i}, f(\widetilde{\mathbf{x}}_{t,i}; \mathbf{w}_t) \right) \right] = \nabla \widetilde{\mathcal{L}}(\mathbf{w}_t), \quad \forall i \in \{1, \ldots, m_0\} \tag{29}$$

and thus

$$\mathrm{E}_{(\widetilde{\mathbf{x}}_t, \widetilde{\mathbf{y}}_t)}[\widetilde{\mathbf{g}}_t] = \nabla \widetilde{\mathcal{L}}(\mathbf{w}_t), \tag{30}$$

where we write $\mathrm{E}_{(\widetilde{\mathbf{x}}_t, \widetilde{\mathbf{y}}_t)}[\widetilde{\mathbf{g}}_t]$ as $\mathrm{E}_{(\widetilde{\mathbf{x}}_{t,1}, \widetilde{\mathbf{y}}_{t,1})}[\ldots \mathrm{E}_{(\widetilde{\mathbf{x}}_{t,m_0}, \widetilde{\mathbf{y}}_{t,m_0})}[\widetilde{\mathbf{g}}_t]]$ for simplicity. Then the norm variance of $\widetilde{\mathbf{g}}_t$ is given by

$$\mathrm{E}_{(\widetilde{\mathbf{x}}_t, \widetilde{\mathbf{y}}_t)}[\|\widetilde{\mathbf{g}}_t - \nabla \widetilde{\mathcal{L}}(\mathbf{w}_t)\|^2]$$

$$= \mathrm{E}_{(\widetilde{\mathbf{x}}_t, \widetilde{\mathbf{y}}_t)} \left[ \left\| \frac{1}{m_0} \sum_{i=1}^{m_0} \nabla \ell \left( \widetilde{\mathbf{y}}_{t,i}, f(\widetilde{\mathbf{x}}_{t,i}; \mathbf{w}_t) \right) - \nabla \widetilde{\mathcal{L}}(\mathbf{w}_t) \right\|^2 \right]$$

$$\overset{(a)}{=} \frac{1}{m_0^2} \sum_{i=1}^{m_0} \mathrm{E}_{(\widetilde{\mathbf{x}}_{t,i}, \widetilde{\mathbf{y}}_{t,i})} \left[ \left\| \nabla \ell \left( \widetilde{\mathbf{y}}_{t,i}, f(\widetilde{\mathbf{x}}_{t,i}; \mathbf{w}_t) \right) - \nabla \widetilde{\mathcal{L}}(\mathbf{w}_t) \right\|^2 \right]$$

$$\overset{(b)}{\leq} \frac{4G^2}{m_0}, \tag{31}$$

where (a) is due to (29) and the pairs $(\widetilde{\mathbf{x}}_{t,1}, \widetilde{\mathbf{y}}_{t,1}), \ldots, (\widetilde{\mathbf{x}}_{t,m_0}, \widetilde{\mathbf{y}}_{t,m_0})$ are independently sampled from $\widetilde{\mathcal{D}}$; (b) is due to the facts that the Assumption of $\widetilde{\mathcal{L}}$ satisfying the property in Definition 1 and Lemma 3, and then by Jensen's inequality, we also have $\|\nabla_\mathbf{w} \widetilde{\mathcal{L}}(\mathbf{w})\| \leq G$, implying that $\left\| \nabla \ell(\widetilde{\mathbf{y}}, f(\widetilde{\mathbf{x}}; \mathbf{w})) - \nabla \widetilde{\mathcal{L}}(\mathbf{w}) \right\|^2 \leq 4G^2$. On the other hand, by the Assumption of $\mathcal{L}$ satisfying the property in Definition 2, we have

$$\mathrm{E}_{(\widetilde{\mathbf{x}}_t, \widetilde{\mathbf{y}}_t)}[\mathcal{L}(\mathbf{w}_{t+1}) - \mathcal{L}(\mathbf{w}_t)]$$

$$\leq \mathrm{E}_{(\widetilde{\mathbf{x}}_t, \widetilde{\mathbf{y}}_t)} \left[ \langle \nabla \mathcal{L}(\mathbf{w}_t), \mathbf{w}_{t+1} - \mathbf{w}_t \rangle + \frac{L}{2} \|\mathbf{w}_{t+1} - \mathbf{w}_t\|^2 \right]$$

$$\overset{(a)}{=} \frac{\eta}{2} \mathrm{E}_{(\widetilde{\mathbf{x}}_t, \widetilde{\mathbf{y}}_t)} \left[ \|\nabla \mathcal{L}(\mathbf{w}_t) - \widetilde{\mathbf{g}}_t\|^2 - \|\nabla \mathcal{L}(\mathbf{w}_t)\|^2 - (1 - \eta L) \|\widetilde{\mathbf{g}}_t\|^2 \right]$$

$$\overset{(b)}{=} \frac{\eta}{2} \left( \|\nabla \mathcal{L}(\mathbf{w}_t) - \nabla \widetilde{\mathcal{L}}(\mathbf{w}_t)\|^2 + \mathrm{E}_{(\widetilde{\mathbf{x}}_t, \widetilde{\mathbf{y}}_t)} \left[ \|\nabla \widetilde{\mathcal{L}}(\mathbf{w}_t) - \widetilde{\mathbf{g}}_t\|^2 \right] - \|\nabla \mathcal{L}(\mathbf{w}_t)\|^2 \right.$$

$$\left. - (1 - \eta L) \mathrm{E}_{(\widetilde{\mathbf{x}}_t, \widetilde{\mathbf{y}}_t)}[\|\widetilde{\mathbf{g}}_t\|^2] \right)$$

$$\overset{(c)}{\leq} \frac{\eta}{2} \left( \|\nabla \mathcal{L}(\mathbf{w}_t) - \nabla \widetilde{\mathcal{L}}(\mathbf{w}_t)\|^2 + \frac{4G^2}{m_0} - \|\nabla \mathcal{L}(\mathbf{w}_t)\|^2 \right) \tag{32}$$

where the (a) is due to the update of $\mathbf{w}_{t+1} = \mathbf{w}_t - \eta \widetilde{\mathbf{g}}_t$; (b) is due to (30); (c) is due to $\eta = 1/L$ and (31). By using the Assumption of $\widetilde{\mathcal{L}}$ and $\mathcal{L}$ satisfying the property in Definition 1 and $\mathbb{P}_\mathbf{x} = \mathbb{P}_{\widetilde{\mathbf{x}}}$, we have

$$\|\nabla \mathcal{L}(\mathbf{w}_t) - \nabla \widetilde{\mathcal{L}}(\mathbf{w}_t)\|$$

$$= \|\mathrm{E}_{(\mathbf{x}, \mathbf{y})}[\nabla_\mathbf{w} \ell(\mathbf{y}, f(\mathbf{x}; \mathbf{w}_t))] - \mathrm{E}_{(\widetilde{\mathbf{x}}, \widetilde{\mathbf{y}})}[\nabla_\mathbf{w} \ell(\widetilde{\mathbf{y}}, f(\widetilde{\mathbf{x}}; \mathbf{w}_t))]\|$$

$$\overset{(a)}{\leq} \mathrm{E}_{(\mathbf{x}, \mathbf{y}, \widetilde{\mathbf{y}})}[\|\nabla_\mathbf{w} \ell(\mathbf{y}, f(\mathbf{x}; \mathbf{w}_t)) - \nabla_\mathbf{w} \ell(\widetilde{\mathbf{y}}, f(\mathbf{x}; \mathbf{w}_t))\|]$$

$$\overset{(21)}{\leq} G \mathrm{E}_{(\mathbf{y}, \widetilde{\mathbf{y}})}[\|\mathbf{y} - \widetilde{\mathbf{y}}\|]$$

$$\overset{(b)}{\leq} G \delta_y, \tag{33}$$

where (a) uses Jensen's inequality; (b) is due to (3). By using the Assumption of $\mathcal{L}$ satisfying the property in Definition 3 and (33), inequality (32) becomes

$$\mathrm{E}_{(\widetilde{\mathbf{x}}_t, \widetilde{\mathbf{y}}_t)}[\mathcal{L}(\mathbf{w}_{t+1}) - \mathcal{L}(\mathbf{w}_t)]$$

$$\leq \frac{\eta G^2 \delta_y^2}{2} + \frac{2\eta G^2}{m_0} - \frac{\eta}{2}\|\nabla \mathcal{L}(\mathbf{w}_t)\|^2$$

$$\leq \frac{\eta G^2 \delta_y^2}{2} + \frac{2\eta G^2}{m_0} - \eta\mu\left(\mathcal{L}(\mathbf{w}_t) - \mathcal{L}(\mathbf{w}_*)\right),$$

which implies

$$\mathrm{E}_{(\widetilde{\mathbf{x}}_t, \widetilde{\mathbf{y}}_t)}[\mathcal{L}(\mathbf{w}_{t+1}) - \mathcal{L}(\mathbf{w}_*)]$$

$$\leq (1 - \eta\mu)\left(\mathcal{L}(\mathbf{w}_t) - \mathcal{L}(\mathbf{w}_*)\right) + \frac{\eta G^2 \delta_y^2}{2} + \frac{2\eta G^2}{m_0}$$

$$\leq (1 - \eta\mu)^t \left(\mathcal{L}(\mathbf{w}_1) - \mathcal{L}(\mathbf{w}_*)\right) + \left(\frac{\eta G^2 \delta_y^2}{2} + \frac{2\eta G^2}{m_0}\right)\sum_{i=0}^{t-1}(1 - \eta\mu)^i.$$

Due to $(1 - \eta\mu)^t \leq \exp(-t\eta\mu)$ and $\sum_{i=0}^{t-1}(1 - \eta\mu)^i \leq \frac{1}{\eta\mu}$, when

$$m_0 \geq \frac{8}{\delta_y^2}$$

and

$$t \geq \frac{L}{\mu}\log\frac{4(\mathcal{L}(\mathbf{w}_1) - \mathcal{L}(\mathbf{w}_*))\mu}{\delta_y^2 G^2},$$

we know

$$\mathrm{E}_{(\widetilde{\mathbf{x}}_t, \widetilde{\mathbf{y}}_t)}[\mathcal{L}(\mathbf{w}_{t+1}) - \mathcal{L}(\mathbf{w}_*)] \leq \frac{\delta_y^2 G^2}{\mu}.$$

$\square$

## D    PROOF OF (9)

We first put the full statement of (9) in the following lemma.

**Lemma 4.** *Assume that $\mathcal{L}$ satisfies the properties in Definition 1, 2 and 3, by setting $\eta = \frac{1}{2n\mu}\log\left(\frac{8n\mu^2(\mathcal{L}(\mathbf{w}_1) - \mathcal{L}(\mathbf{w}_*))}{G^2 L}\right)$, we have $\mathrm{E}_{(\mathbf{x}_n, \mathbf{y}_n)}[\mathcal{L}(\mathbf{w}_{n+1}) - \mathcal{L}(\mathbf{w}_*)] \leq \frac{G^2 L}{8n\mu^2} + \frac{G^2 L}{8n\mu^2}\log\left(\frac{8n\mu^2(\mathcal{L}(\mathbf{w}_1) - \mathcal{L}(\mathbf{w}_*))}{G^2 L}\right)$, where $\mathbf{w}_{n+1}$ is output of SGD trained on $\mathcal{D}$.*

*Proof.* By the Assumption of $\mathcal{L}$ satisfying the property in Definition 2, we have

$$\mathrm{E}_{(\mathbf{x}_n, \mathbf{y}_n)}[\mathcal{L}(\mathbf{w}_{n+1}) - \mathcal{L}(\mathbf{w}_n)]$$

$$\leq \mathrm{E}_{(\mathbf{x}_n, \mathbf{y}_n)}[\langle\nabla\mathcal{L}(\mathbf{w}_n), \mathbf{w}_{n+1} - \mathbf{w}_n\rangle] + \frac{L}{2}\mathrm{E}_{(\mathbf{x}_n, \mathbf{y}_n)}\left[\|\mathbf{w}_{t+n} - \mathbf{w}_n\|^2\right]$$

$$\overset{(a)}{=} -\eta\mathrm{E}_{(\mathbf{x}_n, \mathbf{y}_n)}[\langle\nabla\mathcal{L}(\mathbf{w}_n), \nabla\ell\left(\mathbf{y}_n, f(\mathbf{x}_n; \mathbf{w}_n)\right)\rangle] + \frac{L}{2}\mathrm{E}_{(\mathbf{x}_n, \mathbf{y}_n)}\left[\|\nabla\ell\left(\mathbf{y}_n, f(\mathbf{x}_n; \mathbf{w}_n)\right)\|^2\right]$$

$$\overset{(b)}{=} -\eta\|\nabla\mathcal{L}(\mathbf{w}_n)\|^2 + \frac{\eta^2 L}{2}\mathrm{E}_{(\mathbf{x}_n, \mathbf{y}_n)}[\|\nabla\ell\left(\mathbf{y}_n, f(\mathbf{x}_n; \mathbf{w}_n)\right)\|^2],$$

where (a) is due to the update of $\mathbf{w}_{n+1} = \mathbf{w}_n - \eta\nabla\ell\left(\mathbf{y}_n, f(\mathbf{x}_n; \mathbf{w}_n)\right)$; (b) is due to the Assumption of $\mathcal{L}$ satisfying the property in Definition 1 that $\mathrm{E}_{(\mathbf{x}, \mathbf{y})}\left[\nabla_{\mathbf{w}}\ell\left(\mathbf{y}, f(\mathbf{x}; \mathbf{w})\right)\right] = \nabla\mathcal{L}(\mathbf{w})$. By using the Assumption of $\mathcal{L}$ satisfying the property in Definition 1 that $\|\nabla_{\mathbf{w}}\ell(\mathbf{y}, f(\mathbf{x}; \mathbf{w}))\| \leq G$ and the Assumption of $\mathcal{L}$ satisfying the property in Definition 3, we have

$$\mathrm{E}_{(\mathbf{x}_n, \mathbf{y}_n)}[\mathcal{L}(\mathbf{w}_{n+1}) - \mathcal{L}(\mathbf{w}_n)]$$

$$\leq \frac{\eta^2 L G^2}{2} - \eta\|\nabla\mathcal{L}(\mathbf{w}_n)\|^2$$

$$\leq \frac{\eta^2 L G^2}{2} - 2\eta\mu\left(\mathcal{L}(\mathbf{w}_n) - \mathcal{L}(\mathbf{w}_*)\right),$$

which implies

$$\mathrm{E}_{(\mathbf{x}_n, \mathbf{y}_n)}[\mathcal{L}(\mathbf{w}_{n+1}) - \mathcal{L}(\mathbf{w}_*)]$$

$$\leq (1 - 2\eta\mu) \, \mathrm{E}_{(\mathbf{x}_{n-1}, \mathbf{y}_{n-1})} \left[ \mathcal{L}(\mathbf{w}_n) - \mathcal{L}(\mathbf{w}_*) \right] + \frac{\eta^2 L G^2}{2}$$

$$\leq (1 - 2\eta\mu)^n \left( \mathcal{L}(\mathbf{w}_1) - \mathcal{L}(\mathbf{w}_*) \right) + \frac{\eta^2 L G^2}{2} \sum_{i=0}^{n-1} (1 - 2\eta\mu)^i.$$

Due to $(1 - 2\eta\mu)^n \leq \exp(-2\eta\mu n)$ and $\sum_{i=0}^{n-1}(1 - 2\eta\mu)^i \leq \frac{1}{2\eta\mu}$, then by using the setting of

$$\eta = \frac{1}{2n\mu} \log \left( \frac{8n\mu^2(\mathcal{L}(\mathbf{w}_1) - \mathcal{L}(\mathbf{w}_*))}{G^2 L} \right),$$

we have

$$\mathrm{E}_{(\mathbf{x}_n, \mathbf{y}_n)} \left[ \mathcal{L}(\mathbf{w}_{n+1}) - \mathcal{L}(\mathbf{w}_*) \right]$$

$$\leq \exp\left(-2\eta\mu n\right) \left( \mathcal{L}(\mathbf{w}_1) - \mathcal{L}(\mathbf{w}_*) \right) + \frac{\eta G^2 L}{4\mu}$$

$$= \frac{G^2 L}{8n\mu^2} + \frac{G^2 L}{8n\mu^2} \log \left( \frac{8n\mu^2(\mathcal{L}(\mathbf{w}_1) - \mathcal{L}(\mathbf{w}_*))}{G^2 L} \right)$$

$$\leq O \left( \frac{L}{n\mu^2} \log(n) \right).$$

$\square$

## E  PROOF OF PROPOSITION 1

*Proof.* By using the Assumption of $\widetilde{\mathcal{L}}$ satisfying the property in Definition 3, we have

$$\widetilde{\mathcal{L}}(\mathbf{w}_*) - \widetilde{\mathcal{L}}(\widetilde{\mathbf{w}}_*)$$

$$\leq \frac{\|\nabla\widetilde{\mathcal{L}}(\mathbf{w}_*)\|^2}{2\mu}$$

$$\overset{(a)}{=} \frac{\|\nabla\widetilde{\mathcal{L}}(\mathbf{w}_*) - \nabla\mathcal{L}(\mathbf{w}_*)\|^2}{2\mu}$$

$$\overset{(b)}{\leq} \frac{\delta_y^2 G^2}{2\mu}$$

where (a) is due to the definition of $\mathbf{w}_*$ in (6) so that $\nabla\mathcal{L}(\mathbf{w}_*) = 0$; (b) follows the same analysis of (33) in Lemma 1. Thus we know $\mathbf{w}_* \in \mathcal{A}(\gamma)$ when $\gamma \geq \gamma_0 := \frac{\delta_y^2 G^2}{2\mu}$. On the other hand, by the definition of $\mu(\gamma)$ in (12) and the Assumption of $\mathcal{L}$ satisfying the property in Definition 3, we know $\mu(\gamma) \geq \mu$ when $\gamma \leq 8\mu_0$. $\square$

## F  ALGORITHM AUGDROP AND PROOF OF THEOREM 1

We present the details of update steps for AugDrop and its convergence analysis in this section.

---

**Algorithm 2** AugDrop

---

1: **Input:** $T_1$
2: **Initialize**: $\mathbf{w}_1 \in \mathbb{R}^D, \eta_1, \eta_2 > 0$
    // Stage I: Train Augmented Data
3: **for** $t = 1, 2, \ldots, T_1$ **do**
4:    draw $m_1$ examples $(\widetilde{\mathbf{x}}_{t,1}, \widetilde{\mathbf{y}}_{t,1}), \ldots, (\widetilde{\mathbf{x}}_{t,m_1}, \widetilde{\mathbf{y}}_{t,m_1})$ at random from augmented data
5:    update $\mathbf{w}_{t+1} = \mathbf{w}_t - \frac{\eta_1}{m_1} \sum_{i=1}^{m_1} \nabla_{\mathbf{w}} \ell \left( \widetilde{\mathbf{y}}_{t,i}, f(\widetilde{\mathbf{x}}_{t,i}; \mathbf{w}_t) \right)$
6: **end for**
    // Stage II: Train Original Data
7: **for** $t = T_1 + 1, T_1 + 2, \ldots, T_1 + n/m_2$ **do**
8:    draw $m_2$ examples $(\mathbf{x}_{t,1}, \mathbf{y}_{t,1}), \ldots, (\mathbf{x}_{t,m_2}, \mathbf{y}_{t,m_2})$ without replacement at random from original data
9:    update $\mathbf{w}_{t+1} = \mathbf{w}_t - \frac{\eta_2}{m_2} \sum_{i=1}^{m_2} \nabla_{\mathbf{w}} \ell \left( \mathbf{y}_{t,i}, f(\mathbf{x}_{t,i}; \mathbf{w}_t) \right)$
10: **end for**
11: **Output:** $\mathbf{w}_{T_1 + n/m_2 + 1}$.

---

*Proof.* **In the first stage** of the proposed algorithm, we run a mini-batch SGD over the augmented data $\widetilde{\mathcal{D}}$ with $m_1$ as the size of mini-batch. Let $(\widetilde{\mathbf{x}}_{t,i}, \widetilde{\mathbf{y}}_{t,i}), i = 1, \ldots, m_1$ be the $m_1$ examples sampled in the $t$th iteration. Let $\widetilde{\mathbf{g}}_t$ be the average gradient for the $t$ iteration, i.e.

$$\widetilde{\mathbf{g}}_t = \frac{1}{m_1} \sum_{i=1}^{m_1} \nabla_{\mathbf{w}} \ell(\widetilde{\mathbf{y}}_{t,i}, f(\widetilde{\mathbf{x}}_{t,i}; \mathbf{w}_t))$$

We then update the solution by mini-batch SGD: $\mathbf{w}_{t+1} = \mathbf{w}_t - \eta_1 \widetilde{\mathbf{g}}_t$. By using Lemma 4 of (Ghadimi et al., 2016), with a probability $1 - \delta'$, we have

$$\left\| \widetilde{\mathbf{g}}_t - \nabla \widetilde{\mathcal{L}}(\mathbf{w}_t) \right\| \leq \left( 1 + \sqrt{3 \log \frac{1}{\delta'}} \right) \sqrt{\frac{8G^2}{m_1}}. \tag{34}$$

By the Assumption of $\widetilde{\mathcal{L}}$ satisfying the property in Definition 2 and the update of $\mathbf{w}_{t+1} = \mathbf{w}_t - \eta_1 \widetilde{\mathbf{g}}_t$, we have

$$\widetilde{\mathcal{L}}(\mathbf{w}_{t+1}) - \widetilde{\mathcal{L}}(\mathbf{w}_t)$$
$$\leq - \eta_1 \langle \nabla \widetilde{\mathcal{L}}(\mathbf{w}_t), \widetilde{\mathbf{g}}_t \rangle + \frac{\eta_1^2 L}{2} \|\widetilde{\mathbf{g}}_t\|^2$$
$$= \frac{\eta_1}{2} \|\nabla \widetilde{\mathcal{L}}(\mathbf{w}_t) - \widetilde{\mathbf{g}}_t\|^2 - \frac{\eta_1}{2} \|\nabla \widetilde{\mathcal{L}}(\mathbf{w}_t)\|^2 - \frac{\eta_1 (1 - \eta_1 L)}{2} \|\widetilde{\mathbf{g}}_t\|^2$$
$$\overset{(a)}{\leq} \frac{\eta_1}{2} \left( 1 + \sqrt{3 \log \frac{1}{\delta'}} \right)^2 \frac{8G^2}{m_1} - \eta_1 \mu (\widetilde{\mathcal{L}}(\mathbf{w}_t) - \widetilde{\mathcal{L}}(\widetilde{\mathbf{w}}_*)),$$

where (a) uses the facts that (34), $\eta_1 = 1/L$ and the Assumption of $\widetilde{\mathcal{L}}$ satisfying the property in Definition 3. Thus, with a probability $(1 - \delta')^t$, using the recurrence relation, we have

$$\widetilde{\mathcal{L}}(\mathbf{w}_{t+1}) - \widetilde{\mathcal{L}}(\widetilde{\mathbf{w}}_*)$$
$$\leq (1 - \eta_1 \mu)(\widetilde{\mathcal{L}}(\mathbf{w}_t) - \widetilde{\mathcal{L}}(\widetilde{\mathbf{w}}_*)) + \frac{\eta_1}{2} \left( 1 + \sqrt{3 \log \frac{1}{\delta'}} \right)^2 \frac{8G^2}{m_1}$$
$$\leq (1 - \eta_1 \mu)^t \left( \widetilde{\mathcal{L}}(\mathbf{w}_1) - \widetilde{\mathcal{L}}(\widetilde{\mathbf{w}}_*) \right) + \frac{\eta_1}{2} \left( 1 + \sqrt{3 \log \frac{1}{\delta'}} \right)^2 \frac{8G^2}{m_1} \sum_{i=0}^{t-1} (1 - \eta_1 \mu)^i. \tag{35}$$

Due to $(1 - \eta_1 \mu)^t \leq \exp(-t \eta_1 \mu)$ and $\sum_{i=0}^{t-1} (1 - \eta_1 \mu)^i \leq \frac{1}{\eta_1 \mu}$, when

$$t \geq T_1 := \frac{1}{\eta_1 \mu} \log \frac{2(\widetilde{\mathcal{L}}(\mathbf{w}_1) - \widetilde{\mathcal{L}}(\widetilde{\mathbf{w}}_*))\mu}{\delta_y^2 G^2},$$

we have

$$\widetilde{\mathcal{L}}(\mathbf{w}_{t+1}) - \widetilde{\mathcal{L}}(\widetilde{\mathbf{w}}_*)$$

$$\leq \exp(-t\eta_1\mu)\left(\widetilde{\mathcal{L}}(\mathbf{w}_1) - \widetilde{\mathcal{L}}(\widetilde{\mathbf{w}}_*)\right) + \left(1 + \sqrt{3\log\frac{1}{\delta'}}\right)^2 \frac{4G^2}{\mu m_1}$$

$$\leq \frac{\delta_y^2 G^2}{2\mu} + \left(1 + \sqrt{3\log\frac{1}{\delta'}}\right)^2 \frac{4G^2}{\mu m_1}. \tag{36}$$

Let $\delta' = \frac{\delta}{2T_1}$, if we choose $m_1$ such that

$$m_1 = \left(1 + \sqrt{3\log\frac{2T_1}{\delta}}\right)^2 \frac{8}{\delta_y^2},$$

then for any $t \geq T_1$, then with a probability $1 - \delta/2$ we have

$$\widetilde{\mathcal{L}}(\mathbf{w}_{t+1}) - \widetilde{\mathcal{L}}(\widetilde{\mathbf{w}}_*) \leq \frac{\delta_y^2 G^2}{\mu}. \tag{37}$$

**In the second stage** of the proposed algorithm, we run a mini-batch SGD over the original data $\mathcal{D}$ with $m_2$ as the size of mini-batch. Let $(\mathbf{x}_{t,i}, \mathbf{y}_{t,i}), i = 1, \ldots, m_2$ be the $m_2$ examples sampled in the $t$th iteration. Let $\widehat{\mathbf{g}}_t$ be the average gradient for the $t$ iteration, i.e.

$$\widehat{\mathbf{g}}_t = \frac{1}{m_2} \sum_{i=1}^{m_2} \nabla_{\mathbf{w}} \ell(\mathbf{y}_{t,i}, f(\mathbf{x}_{t,i}; \mathbf{w}_t))$$

We then update the solution $\mathbf{w}_{t+1} = \mathbf{w}_t - \eta_2 \widehat{\mathbf{g}}_t$. By using Lemma 4 of (Ghadimi et al., 2016), with a probability $1 - \delta''$, we have

$$\|\widehat{\mathbf{g}}_t - \nabla\mathcal{L}(\mathbf{w}_t)\| \leq \left(1 + \sqrt{3\log\frac{1}{\delta''}}\right)\sqrt{\frac{8G^2}{m_2}}. \tag{38}$$

By the smoothness of $\widetilde{\mathcal{L}}(\mathbf{w})$ and the update of $\mathbf{w}_{t+1} = \mathbf{w}_t - \eta_2 \widehat{\mathbf{g}}_t$, we have

$$\widetilde{\mathcal{L}}(\mathbf{w}_{t+1}) - \widetilde{\mathcal{L}}(\mathbf{w}_t)$$

$$\leq -\eta_2\langle\nabla\widetilde{\mathcal{L}}(\mathbf{w}_t), \widehat{\mathbf{g}}_t\rangle + \frac{\eta_2^2 L}{2}\|\widehat{\mathbf{g}}_t\|^2$$

$$= \frac{\eta_2}{2}\|\nabla\widetilde{\mathcal{L}}(\mathbf{w}_t) - \widehat{\mathbf{g}}_t\|^2 - \frac{\eta_2}{2}\|\nabla\widetilde{\mathcal{L}}(\mathbf{w}_t)\|^2 - \frac{\eta_2(1 - \eta_2 L)}{2}\|\widehat{\mathbf{g}}_t\|^2$$

$$\overset{(a)}{\leq} \eta_2\|\nabla\widetilde{\mathcal{L}}(\mathbf{w}_t) - \nabla\mathcal{L}(\mathbf{w}_t)\|^2 + \eta_2\|\widehat{\mathbf{g}}_t - \nabla\mathcal{L}(\mathbf{w}_t)\|^2 - \eta_2\mu(\widetilde{\mathcal{L}}(\mathbf{w}_t) - \widetilde{\mathcal{L}}(\widetilde{\mathbf{w}}_*))$$

$$\overset{(b)}{\leq} \eta_2\left(2\delta_y^2 G^2 + \left(1 + \sqrt{3\log\frac{1}{\delta''}}\right)^2 \frac{8G^2}{m_2}\right) - \eta_2\mu(\widetilde{\mathcal{L}}(\mathbf{w}_t) - \widetilde{\mathcal{L}}(\widetilde{\mathbf{w}}_*)), \tag{39}$$

where (a) uses the facts that Young's inequality, $\eta_2 \leq 1/L$ and the Assumption of $\widetilde{\mathcal{L}}$ satisfying the property in Definition 3; (b) uses inequality (38) and the same analysis of (33) in Lemma 1. It is easy to verify that for any $t \in \{T_1 + 1, \ldots, T_1 + n/m_2\}$, we have, with a probability $(1 - \delta'')^{n/m_2}$, we have

$$\widetilde{\mathcal{L}}(\mathbf{w}_{t+1}) - \widetilde{\mathcal{L}}(\widetilde{\mathbf{w}}_*)$$

$$\leq (1 - \eta_2\mu)\left(\widetilde{\mathcal{L}}(\mathbf{w}_t) - \widetilde{\mathcal{L}}(\widetilde{\mathbf{w}}_*)\right) + \eta_2\left(G^2\delta_y^2 + \left(1 + \sqrt{3\log\frac{1}{\delta''}}\right)^2 \frac{8G^2}{m_2}\right)$$

$$\leq (1 - \eta_2\mu)^t\left(\widetilde{\mathcal{L}}(\mathbf{w}_{T_1+1}) - \widetilde{\mathcal{L}}(\widetilde{\mathbf{w}}_*)\right) + \eta_2\left(G^2\delta_y^2 + \left(1 + \sqrt{3\log\frac{1}{\delta''}}\right)^2 \frac{8G^2}{m_2}\right)\sum_{i=0}^{t-1}(1 - \eta_2\mu)^i$$

$$\leq \frac{\delta_y^2 G^2}{\mu} + \frac{1}{\mu}\left(G^2\delta_y^2 + \left(1 + \sqrt{3\log\frac{1}{\delta''}}\right)^2 \frac{8G^2}{m_2}\right),$$

where the last inequality is due to $(1 - \eta_2\mu)^t \leq 1$, $\sum_{i=0}^{t-1}(1 - \eta_2\mu)^i \leq \frac{1}{\eta_2\mu}$, and (37). Let $\delta'' = \frac{\delta}{2n/m_2}$, if we choose $m_2$ such that $m_2 \geq \left(1 + \sqrt{3\log\frac{2n}{m_2\delta}}\right)^2 \frac{4}{\delta_y^2}$, for example,

$$m_2 = \left(1 + \sqrt{3\log\frac{2n}{\delta}}\right)^2 \frac{4}{\delta_y^2},$$

then with a probability $1 - \delta$, for any $t \in \{T_1 + 1, \ldots, T_1 + n/m_2\}$ we have

$$\widetilde{\mathcal{L}}(\mathbf{w}_{t+1}) - \widetilde{\mathcal{L}}(\widetilde{\mathbf{w}}_*) \leq \frac{4\delta_y^2 G^2}{\mu}.$$

Therefore, $\mathbf{w}_t \in \mathcal{A}(8\gamma_0)$ for any $t \in \{T_1 + 2, \ldots, T_1 + n/m_2 + 1\}$. Following the standard analysis in Appendix D, we have

$$\mathrm{E}\left[\mathcal{L}(\mathbf{w}_{T_1+n/m_2+1}) - \mathcal{L}(\mathbf{w}_*)\right] \leq \frac{G^2 L}{4n\mu_c^2} + \frac{G^2 L}{4n\mu_c^2}\log\left(\frac{4n\mu_c^2(\mathcal{L}(\mathbf{w}_1) - \mathcal{L}(\mathbf{w}_*))}{G^2 L}\right),$$

where $\mu_c = \mu(8\gamma_0)$. $\qquad\square$

## G  OPTIMAL SOLUTIONS OF $\mathcal{L}_a(\mathbf{w})$ AND $\mathcal{L}_c(\mathbf{w})$

By the definition of $\ell_a$ in (14) and $\mathbb{P}_\mathbf{x} = \mathbb{P}_{\widetilde{\mathbf{x}}}$, we know

$$\begin{aligned}
&\ell_a(\widetilde{\mathbf{y}}, f(\widetilde{\mathbf{x}}; \mathbf{w})) \\
&= \min_{\|\mathbf{z}-\widetilde{\mathbf{y}}\| \leq \delta_y} \ell(\mathbf{z}, f(\mathbf{x}; \mathbf{w})) \\
&\leq \ell(\mathbf{y}, f(\mathbf{x}; \mathbf{w}))
\end{aligned} \tag{40}$$

since $\|\mathbf{y} - \widetilde{\mathbf{y}}\| \leq \delta_y$. Therefore, by (15), (40) and $\mathbb{P}_\mathbf{x} = \mathbb{P}_{\widetilde{\mathbf{x}}}$ we have

$$\begin{aligned}
&\mathcal{L}_a(\mathbf{w}) \\
&= \mathrm{E}_\mathbf{y}[\mathcal{L}_a(\mathbf{w})] \\
&= \mathrm{E}_{(\mathbf{x},\widetilde{\mathbf{y}},\mathbf{y})}\left[\ell_a(\widetilde{\mathbf{y}}, f(\mathbf{x}; \mathbf{w}))\right] \\
&\leq \mathrm{E}_{(\mathbf{x},\widetilde{\mathbf{y}},\mathbf{y})}\left[\ell(\mathbf{y}, f(\mathbf{x}; \mathbf{w}))\right] \\
&= \mathrm{E}_{(\mathbf{x},\mathbf{y})}\left[\ell(\mathbf{y}, f(\mathbf{x}; \mathbf{w}))\right] \\
&= \mathcal{L}(\mathbf{w}).
\end{aligned} \tag{41}$$

Since $\ell$ is a non-negative loss function, then we know

$$0 \leq \mathcal{L}_a(\mathbf{w}_*) \leq \mathcal{L}(\mathbf{w}_*) = 0,$$

which implies that

$$\mathcal{L}_a(\mathbf{w}_*) = 0,$$

and thus

$$\mathcal{L}_a(\mathbf{w}_*) \leq \mathcal{L}_a(\mathbf{w}), \quad \forall \mathbf{w}.$$

Therefore, $\mathbf{w}_*$ also minimizes $\mathcal{L}_a(\mathbf{w})$.

On the other hand, by (16) we know

$$\mathcal{L}_c(\mathbf{w}_*) = \lambda\mathcal{L}(\mathbf{w}_*) + (1 - \lambda)\mathcal{L}_a(\mathbf{w}_*) = 0.$$

Therefore,

$$\mathcal{L}_c(\mathbf{w}_*) \leq \mathcal{L}_c(\mathbf{w}), \quad \forall \mathbf{w},$$

i.e, $\mathbf{w}_*$ also minimizes $\mathcal{L}_c(\mathbf{w})$, indicating that $\mathcal{L}_c(\mathbf{w})$ shares the same minimizer as $\mathcal{L}(\mathbf{w})$.

# H    ALGORITHM MIXLOSS AND PROOF OF THEOREM 2

We present the details of update steps for MixLoss and its convergence analysis in this section.

---

**Algorithm 3** MixLoss

---

1: **Input:** $\lambda$
2: **Initialize**: $\mathbf{w}_1 \in \mathbb{R}^D, \eta > 0$
3: **for** $t = 1, 2, \ldots, n$ **do**
4:    draw an example $(\mathbf{x}_t, \mathbf{y}_t)$ without replacement at random from original data
5:    draw $m_0$ examples $(\widetilde{\mathbf{x}}_{t,1}, \widetilde{\mathbf{y}}_{t,1}), \ldots, (\widetilde{\mathbf{x}}_{t,m_0}, \widetilde{\mathbf{y}}_{t,m_0})$ at random from augmented data
6:    compute $\widehat{\mathbf{g}}_t = \lambda \nabla \ell(\mathbf{y}_t, f(\mathbf{x}_t; \mathbf{w}_t)) + (1 - \lambda)\frac{1}{m_0} \sum_{i=1}^{m_0} \nabla \ell_a(\widetilde{\mathbf{y}}_{t,i}, f(\widetilde{\mathbf{x}}_{t,i}; \mathbf{w}_t))$
7:    update $\mathbf{w}_{t+1} = \mathbf{w}_t - \eta \widehat{\mathbf{g}}_t$
8: **end for**
9: **Output:** $\mathbf{w}_{n+1}$.

---

*Proof.* Recall that

$$\mathcal{L}_c(\mathbf{w}) = \lambda \mathcal{L}(\mathbf{w}) + (1 - \lambda)\mathcal{L}_a(\mathbf{w}), \tag{42}$$

where $\mathcal{L}_a(\mathbf{w}) = \mathrm{E}_{(\widetilde{\mathbf{x}}, \widetilde{\mathbf{y}})} \left[ \ell_a(\widetilde{\mathbf{y}}, f(\widetilde{\mathbf{x}}; \mathbf{w})) \right] = \mathrm{E}_{(\widetilde{\mathbf{x}}, \widetilde{\mathbf{y}})} \left[ \min_{\|\mathbf{z} - \widetilde{\mathbf{y}}\| \le \delta_y} \ell(\mathbf{z}, f(\widetilde{\mathbf{x}}; \mathbf{w})) \right]$ and

$$\widehat{\mathbf{g}}_t = \lambda \nabla \ell(\mathbf{y}_t, f(\mathbf{x}_t; \mathbf{w}_t)) + (1 - \lambda)\frac{1}{m_0} \sum_{i=1}^{m_0} \nabla \ell_a(\widetilde{\mathbf{y}}_{t,i}, f(\widetilde{\mathbf{x}}_{t,i}; \mathbf{w}_t)). \tag{43}$$

By the update of $\mathbf{w}_{t+1} = \mathbf{w}_t - \eta \widehat{\mathbf{g}}_t$ and by the Assumption of $\mathcal{L}_c$ satisfying the property in Definition 2, we have (Nesterov, 2004)

$$\mathrm{E}_t \left[ \mathcal{L}_c(\mathbf{w}_{t+1}) - \mathcal{L}_c(\mathbf{w}_t) \right]$$

$$\le -\eta \mathrm{E}_t \left[ \langle \nabla \mathcal{L}_c(\mathbf{w}_t), \widehat{\mathbf{g}}_t \rangle \right] + \frac{\eta^2 L}{2} \mathrm{E}_t \left[ \|\widehat{\mathbf{g}}_t\|^2 \right]$$

$$\overset{(a)}{\le} -\eta(1 - \eta L)\mathrm{E}_t \left[ \|\nabla \mathcal{L}_c(\mathbf{w}_t)\|^2 \right] + \eta^2 L \mathrm{E}_t \left[ \|\widehat{\mathbf{g}}_t - \nabla \mathcal{L}_c(\mathbf{w}_t)\|^2 \right]$$

$$\overset{(b)}{\le} -\eta(1 - \eta L)\mathrm{E}_t \left[ \|\nabla \mathcal{L}_c(\mathbf{w}_t)\|^2 \right] + \frac{9}{8}\lambda^2 \eta^2 L \mathrm{E}_t \left[ \|\nabla \ell(\mathbf{y}_t, f(\mathbf{x}_t; \mathbf{w}_t)) - \nabla \mathcal{L}(\mathbf{w}_t)\|^2 \right]$$

$$+ 9(1 - \lambda)^2 \eta^2 L \mathrm{E}_t \left[ \left\| \frac{1}{m_0} \sum_{i=1}^{m_0} \nabla \ell_a(\widetilde{\mathbf{y}}_{t,i}, f(\widetilde{\mathbf{x}}_{t,i}; \mathbf{w}_t)) - \nabla \mathcal{L}_a(\mathbf{w}_t) \right\|^2 \right]$$

$$\overset{(c)}{\le} -\eta(1 - \eta L)\mathrm{E}_t \left[ \|\nabla \mathcal{L}_c(\mathbf{w}_t)\|^2 \right] + \frac{9}{2}\lambda^2 \eta^2 L G^2 + \frac{36(1 - \lambda)^2 \eta^2 L G^2}{m_0}$$

$$\overset{(d)}{\le} -\eta(1 - \eta L)\mathrm{E}_t \left[ \|\nabla \mathcal{L}_c(\mathbf{w}_t)\|^2 \right] + 5\lambda^2 \eta^2 L G^2, \tag{44}$$

where $\mathrm{E}_t[\cdot]$ is taken over random variables $(\mathbf{x}_t, \mathbf{y}_t), (\widetilde{\mathbf{x}}_{t,1}, \widetilde{\mathbf{y}}_{t,1}), \ldots, (\widetilde{\mathbf{x}}_{t,m_0}, \widetilde{\mathbf{y}}_{t,m_0})$; (a) uses the facts that Young's inequality $\|\mathbf{a} - \mathbf{b}\|^2 \le 2\|\mathbf{a}\|^2 + 2\|\mathbf{b}\|^2$ and $\mathrm{E}[\widehat{\mathbf{g}}_t] = \nabla \mathcal{L}_c(\mathbf{w}_t)$; (b) uses the facts that (42) (43) and Young's inequality $\|\mathbf{a} + \mathbf{b}\|^2 \le (1 + 1/c)\|\mathbf{a}\|^2 + (1 + c)\|\mathbf{b}\|^2$ with $a = 8$; (c) use the same analysis in (31) from the proof of Lemma 1, the facts that the Assumption of $\mathcal{L}$ satisfying the property in Definition 1 and by Jensen's inequality, we also have $\|\nabla \mathcal{L}(\mathbf{w})\| \le G$, implying that $\|\nabla \ell(\mathbf{y}, f(\mathbf{x}; \mathbf{w})) - \nabla \mathcal{L}(\mathbf{w})\|^2 \le 4G^2$; (d) holds by setting $m_0 \ge \frac{72(1-\lambda)^2}{\lambda^2}$ since we have sufficiently large number of augmented examples. Thus, since $\eta \le \frac{1}{2L}$ and by using the Assumption of $\mathcal{L}_c$ satisfying the property in Definition 3, we have

$$\mathrm{E}_t \left[ \mathcal{L}_c(\mathbf{w}_{t+1}) - \mathcal{L}_c(\mathbf{w}_t) \right] \le -\eta \mu \mathrm{E}_t \left[ \mathcal{L}_c(\mathbf{w}_t) \right] + 5\lambda^2 \eta^2 L G^2,$$

and therefore

$$
\mathrm{E}_n \left[ \mathcal{L}_c(\mathbf{w}_{n+1}) \right]
$$
$$
\leq \exp\left(-\eta\mu n\right) \mathcal{L}_c(\mathbf{w}_1) + \frac{5\lambda^2 \eta L G^2}{\mu}
$$
$$
\leq \exp\left(-\eta\mu n\right) \mathcal{L}(\mathbf{w}_1) + \frac{5\lambda^2 \eta L G^2}{\mu},
\tag{45}
$$

where last inequality is due to the fact that $\mathcal{L}_c(\mathbf{w}) \leq \mathcal{L}(\mathbf{w})$. In (45), by choosing

$$
\eta = \frac{1}{\mu n} \log \frac{n \mu^2 \mathcal{L}(\mathbf{w}_1)}{\lambda^2 L G^2},
$$

we have

$$
\mathrm{E}_n \left[ \mathcal{L}_c(\mathbf{w}_{n+1}) \right] \leq \frac{\lambda^2 L G^2}{n \mu^2} + \frac{5\lambda^2 L G^2}{n \mu^2} \log \frac{n \mu^2 \mathcal{L}(\mathbf{w}_1)}{\lambda^2 L G^2}.
\tag{46}
$$

Since $\mathcal{L}(\mathbf{w}) = \frac{1}{\lambda}\mathcal{L}_c(\mathbf{w}) - \frac{1-\lambda}{\lambda}\mathcal{L}_a(\mathbf{w})$, then (46) becomes

$$
\mathrm{E}_n \left[ \mathcal{L}(\mathbf{w}_{n+1}) \right]
$$
$$
\leq \frac{\lambda L G^2}{n \mu^2} + \frac{5\lambda L G^2}{n \mu^2} \log \frac{n \mu^2 \mathcal{L}(\mathbf{w}_1)}{\lambda^2 L G^2} - \frac{1-\lambda}{\lambda} \mathrm{E}\left[\mathcal{L}_a(\mathbf{w}_{n+1})\right]
$$
$$
\leq \frac{\lambda L G^2}{n \mu^2} + \frac{5\lambda L G^2}{n \mu^2} \log \frac{n \mu^2 \mathcal{L}(\mathbf{w}_1)}{\lambda^2 L G^2},
\tag{47}
$$

where the last inequality is due to $\lambda \in (0,1)$ and $\mathcal{L}_a(\mathbf{w}_{n+1}) \geq \mathcal{L}_a(\mathbf{w}_*) = 0$. $\qquad\square$

# I   PROOFS IN APPENDIX A

We include the proofs for Appendix section "Main Results for label-preserving Augmentation".

## I.1   PROOF OF LEMMA 2

The analysis is similar to that for Lemma 1. For completeness, we include it here.

*Proof.* Following the same analysis in Lemma 1, we can have the same result as in (32). That is to say, we have

$$
\mathrm{E}_{(\widetilde{\mathbf{x}}_t, \widetilde{\mathbf{y}}_t)}[\mathcal{L}(\mathbf{w}_{t+1}) - \mathcal{L}(\mathbf{w}_t)] \leq \frac{\eta}{2} \left( \|\nabla\mathcal{L}(\mathbf{w}_t) - \nabla\widetilde{\mathcal{L}}(\mathbf{w}_t)\|^2 + \frac{4G^2}{m_0} - \|\nabla\mathcal{L}(\mathbf{w}_t)\|^2 \right).
\tag{48}
$$

We have

$$
\|\nabla\mathcal{L}(\mathbf{w}_t) - \nabla\widetilde{\mathcal{L}}(\mathbf{w}_t)\|
$$
$$
\leq \int d\mathbf{x}\mathbf{y} \|\mathbb{P}_{\mathbf{x}}(\mathbf{x}) - \mathbb{P}_{\widetilde{\mathbf{x}}}(\mathbf{x})\| \|\nabla\ell(\mathbf{y}, f(\mathbf{x}; \mathbf{w}_t))\|
$$
$$
\overset{(a)}{\leq} G \int d\mathbf{x} \|\mathbb{P}_{\mathbf{x}}(\mathbf{x}) - \mathbb{P}_{\widetilde{\mathbf{x}}}(\mathbf{x})\|
$$
$$
\overset{(b)}{\leq} G \sqrt{2 D_{KL}(\mathbb{P}_{\mathbf{x}} \| \mathbb{P}_{\widetilde{\mathbf{x}}})}
$$
$$
\overset{(c)}{=} G \sqrt{2 \delta_P},
\tag{49}
$$

where (a) is due to the Assumption of $\mathcal{L}$ satisfying the property in Definition 1; (b) uses Pinsker's inequality (Csiszar & Körner, 2011; Tsybakov, 2008); (c) is due to (18). With inequality (49),

by using the facts that $\eta = 1/L$ and the Assumption of $\mathcal{L}$ satisfying the property in Definition 3, inequality (48) becomes

$$\mathrm{E}_{(\widetilde{\mathbf{x}}_t, \widetilde{\mathbf{y}}_t)}[\mathcal{L}(\mathbf{w}_{t+1}) - \mathcal{L}(\mathbf{w}_t)]$$

$$\leq \eta \delta_P G^2 + \frac{4G^2}{m_0} - \frac{\eta}{2} \|\nabla \mathcal{L}(\mathbf{w}_t)\|^2$$

$$\leq \eta \delta_P G^2 + \frac{4G^2}{m_0} - \eta \mu \left( \mathcal{L}(\mathbf{w}_t) - \mathcal{L}(\mathbf{w}_*) \right)$$

$$\leq 2\eta \delta_P G^2 - \eta \mu \left( \mathcal{L}(\mathbf{w}_t) - \mathcal{L}(\mathbf{w}_*) \right),$$

where the last inequality is due to the selection of $m_0 \geq \frac{4}{\eta \delta_P}$. Then we have

$$\mathrm{E}_{(\widetilde{\mathbf{x}}_t, \widetilde{\mathbf{y}}_t)}[\mathcal{L}(\mathbf{w}_{t+1}) - \mathcal{L}(\mathbf{w}_*)]$$

$$\leq (1 - \eta\mu) \, \mathrm{E}_{(\widetilde{\mathbf{x}}_{t-1}, \widetilde{\mathbf{y}}_{t-1})}[\mathcal{L}(\mathbf{w}_t) - \mathcal{L}(\mathbf{w}_*)] + 2\eta \delta_P G^2$$

$$\leq (1 - \eta\mu)^t \left( \mathcal{L}(\mathbf{w}_1) - \mathcal{L}(\mathbf{w}_*) \right) + 2\eta \delta_P G^2 \sum_{i=0}^{t-1} (1 - \eta\mu)^i.$$

Due to $(1 - \eta\mu)^t \leq \exp(-t\eta\mu)$ and $\sum_{i=0}^{t-1}(1 - \eta\mu)^i \leq \frac{1}{\eta\mu}$, when

$$t \geq \frac{L}{\mu} \log \frac{(\mathcal{L}(\mathbf{w}_1) - \mathcal{L}(\mathbf{w}_*))\mu}{2\delta_P G^2},$$

we know

$$\mathrm{E}_{(\widetilde{\mathbf{x}}_t, \widetilde{\mathbf{y}}_t)}[\mathcal{L}(\mathbf{w}_{t+1}) - \mathcal{L}(\mathbf{w}_*)] \leq \frac{4\delta_P G^2}{\mu}.$$

$\square$

## I.2    PROOF OF PROPOSITION 2

*Proof.* By using the Assumption of $\widetilde{\mathcal{L}}$ satisfying the property in Definition 3, we have

$$\widetilde{\mathcal{L}}(\mathbf{w}_*) - \widetilde{\mathcal{L}}(\widetilde{\mathbf{w}}_*)$$

$$\leq \frac{\|\nabla \widetilde{\mathcal{L}}(\mathbf{w}_*)\|^2}{2\mu}$$

$$\overset{(a)}{=} \frac{\|\nabla \widetilde{\mathcal{L}}(\mathbf{w}_*) - \nabla \mathcal{L}(\mathbf{w}_*)\|^2}{2\mu}$$

$$\overset{(b)}{\leq} \frac{\delta_P G^2}{\mu}$$

where (a) is due to the definition of $\mathbf{w}_*$ in (6) so that $\nabla \mathcal{L}(\mathbf{w}_*) = 0$; (b) follows the same analysis of (49) in Lemma 2. Thus we know $\mathbf{w}_* \in \mathcal{A}(\gamma)$ when $\gamma \geq \gamma_1 := \frac{\delta_P G^2}{\mu}$. On the other hand, by the definition of $\mu(\gamma)$ in (12) and the Assumption of $\mathcal{L}$ satisfying the property in Definition 3, we know $\mu(\gamma) \geq \mu$ when $\gamma \leq 4\mu_1$.

$\square$

## I.3    PROOF OF THEOREM 3

This proof is similar to the proof of Theorem 1. For completeness, we include it here.

*Proof.* **In the first stage** of the proposed algorithm, we run a mini-batch SGD over the augmented data $\widetilde{\mathcal{D}}$ with $m_1$ as the size of mini-batch. Using the similar analysis in (35) from Theorem 1, we have, with a probability $(1 - \delta')^t$,

$$\widetilde{\mathcal{L}}(\mathbf{w}_{t+1}) - \widetilde{\mathcal{L}}(\widetilde{\mathbf{w}}_*) \leq (1 - \eta_1\mu)^t \left( \widetilde{\mathcal{L}}(\mathbf{w}_1) - \widetilde{\mathcal{L}}(\widetilde{\mathbf{w}}_*) \right) + \frac{\eta_1}{2} \left( 1 + \sqrt{3 \log \frac{1}{\delta'}} \right)^2 \frac{8G^2}{m_1} \sum_{i=0}^{t-1} (1 - \eta_1\mu)^i.$$

Due to $(1 - \eta_1\mu)^t \leq \exp(-t\eta_1\mu)$ and $\sum_{i=0}^{t-1}(1 - \eta_1\mu)^i \leq \frac{1}{\eta_1\mu}$, when

$$t \geq T_1 := \frac{L}{\mu}\log\frac{2(\widetilde{\mathcal{L}}(\mathbf{w}_1) - \widetilde{\mathcal{L}}(\widetilde{\mathbf{w}}_*))\mu}{\delta_P G^2},$$

we have

$$\widetilde{\mathcal{L}}(\mathbf{w}_{t+1}) - \widetilde{\mathcal{L}}(\widetilde{\mathbf{w}}_*) \leq \frac{\delta_P G^2}{2\mu} + \left(1 + \sqrt{3\log\frac{1}{\delta'}}\right)^2 \frac{8G^2}{2\mu m_1}. \tag{50}$$

Let $\delta' = \frac{\delta}{2T_1}$, if we choose $m_1$ such that

$$m_1 = \left(1 + \sqrt{3\log\frac{2T_1}{\delta}}\right)^2 \frac{8}{\delta_P},$$

then for any $t \geq T_1$, we have

$$\widetilde{\mathcal{L}}(\mathbf{w}_{t+1}) - \widetilde{\mathcal{L}}(\widetilde{\mathbf{w}}_*) \leq \frac{\delta_P G^2}{\mu}. \tag{51}$$

**In the second stage** of the proposed algorithm, we run a mini-batch SGD over the original data $\mathcal{D}$ with $m_2$ as the size of mini-batch. Using the same analysis in (39) from Theorem 1, we have

$$\widetilde{\mathcal{L}}(\mathbf{w}_{t+1}) - \widetilde{\mathcal{L}}(\mathbf{w}_t) \leq \eta_2\|\nabla\widetilde{\mathcal{L}}(\mathbf{w}_t) - \nabla\mathcal{L}(\mathbf{w}_t)\|^2 + \eta_2\|\widehat{\mathbf{g}}_t - \nabla\mathcal{L}(\mathbf{w}_t)\|^2 - \eta_2\mu(\widetilde{\mathcal{L}}(\mathbf{w}_t) - \widetilde{\mathcal{L}}(\widetilde{\mathbf{w}}_*)).$$

Then by (38) in the proof of Theorem 1 and (49) in the proof of Lemma 2, we have, with a probability $1 - \delta''$,

$$\widetilde{\mathcal{L}}(\mathbf{w}_{t+1}) - \widetilde{\mathcal{L}}(\mathbf{w}_t) \leq 2\eta_2\delta_P G^2 + \eta_2\left(1 + \sqrt{3\log\frac{1}{\delta''}}\right)^2 \frac{8G^2}{m_2} - \eta_2\mu(\widetilde{\mathcal{L}}(\mathbf{w}_t) - \widetilde{\mathcal{L}}(\widetilde{\mathbf{w}}_*)).$$

It is easy to verify that for any $t \in \{T_1 + 1, \ldots, T_1 + n/m_2\}$, we have, with a probability $(1 - \delta'')^{n/m_2}$, we have

$$\widetilde{\mathcal{L}}(\mathbf{w}_{t+1}) - \widetilde{\mathcal{L}}(\widetilde{\mathbf{w}}_*)$$

$$\leq (1 - \eta_2\mu)\left(\widetilde{\mathcal{L}}(\mathbf{w}_t) - \widetilde{\mathcal{L}}(\widetilde{\mathbf{w}}_*)\right) + \eta_2\left(2\delta_P G^2 + \left(1 + \sqrt{3\log\frac{1}{\delta''}}\right)^2 \frac{8G^2}{m_2}\right)$$

$$\leq (1 - \eta_2\mu)^{n/m_2}\left(\widetilde{\mathcal{L}}(\mathbf{w}_{T_1+1}) - \widetilde{\mathcal{L}}(\widetilde{\mathbf{w}}_*)\right) + \eta_2\left(2\delta_P G^2 + \left(1 + \sqrt{3\log\frac{1}{\delta''}}\right)^2 \frac{8G^2}{m_2}\right)\sum_{i=0}^{n/m_2-1}(1 - \eta_2\mu)^i$$

$$\leq \frac{\delta_P G^2}{\mu} + \frac{1}{\mu}\left(2\delta_P G^2 + \left(1 + \sqrt{3\log\frac{1}{\delta''}}\right)^2 \frac{8G^2}{m_2}\right),$$

where the last inequality is due to $(1 - \eta_2\mu)^t \leq 1$, $\sum_{i=0}^{t-1}(1 - \eta_2\mu)^i \leq \frac{1}{\eta_2\mu}$, and (51). Let $\delta'' = \frac{\delta}{2n/m_2}$, if we choose $m_2$ such that $m_2 \geq \left(1 + \sqrt{3\log\frac{2n}{m_2\delta}}\right)^2 \frac{8}{\delta_P}$, for example,

$$m_2 = \left(1 + \sqrt{3\log\frac{2n}{\delta}}\right)^2 \frac{8}{\delta_P},$$

then with a probability $1 - \delta$, for any $t \in \{T_1 + 1, \ldots, T_1 + n/m_2\}$ we have

$$\widetilde{\mathcal{L}}(\mathbf{w}_{t+1}) - \widetilde{\mathcal{L}}(\widetilde{\mathbf{w}}_*) \leq \frac{4\delta_P G^2}{\mu}.$$

Therefore, $\mathbf{w}_t \in \mathcal{A}(4\gamma_1)$ for any $t \in \{T_1 + 2, \ldots, T_1 + n/m_2 + 1\}$. Following the similar analysis in Appendix D, we have

$$\mathrm{E}\left[\mathcal{L}(\mathbf{w}_{T_1+n/m_2+1}) - \mathcal{L}(\mathbf{w}_*)\right] \leq \frac{G^2 L}{4n\mu_e^2} + \frac{G^2 L}{4n\mu_e^2}\log\left(\frac{4n\mu_e^2(\mathcal{L}(\mathbf{w}_1) - \mathcal{L}(\mathbf{w}_*))}{G^2 L}\right),$$

where $\mu_e = \mu(4\gamma_1)$. $\qquad\square$

