# OpenReview forum: "WeMix: How to Better Utilize Data Augmentation"
_ICLR.cc/2021/Conference — Reject_

### Official Review · AnonReviewer1 · 2020-10-28
**Interesting algorithms + theoretical analysis but weak experimental validation**

**Rating:** 4
**Confidence:** 3

**Review:**

This paper theoretically analyzes the effect of bias in augmented data on the efficacy of data augmentation approaches, and proposes three new approaches to leveraging data augmentation in ways that purport to limit the negative effects of data bias in the augmented data distribution, and both theoretically and experimentally analyze them.  The algorithmic approaches and analyses are interesting, but (A) the theoretical results don't deliver a lot of specific actionable insight beyond the high level intuition they start with, and (B) the experimental section is somewhat weak.

In the main analysis (Sec. 4), the authors establish a bound (Lemma 1) on the generalization error due to training on augmented data (nit: unclear why $w_{t+1}$ is used to denote the final (?) params learned...?) and compare with a standard bound training on normal data.  However, the bound in Lemma 1 is not overly helpful (quadratic dependence on max augmented label divergence- unclear how this helps / is actionable), and the comparison with the normal bound doesn't reveal much but the original intuition that this approach *could* easily introduce bias, as we started with.

The paper then proposes two algorithms- "AugDrop", which uses a constrained optimization approach, and a corresponding theoretical analysis that shows potentially less biased and better performance than training on normal augmented data- and "MixLoss".  They also propose combining the two into "WeMix".

At this point, the main issue is with the experiments.  The authors could potentially have tried to fulfill two main objectives here: (A) make this mostly a theory paper, and seek to explore the quantities and tradeoffs of interest in the theoretical analysis in some experiments; and/or (B) show the practical efficacy of the proposed algorithms by running them with real data augmentation approaches, compared against modern approaches.  However, the experiments section accomplishes neither.
- Re (A): There is no detailed experimental analysis beyond comparing end performance of models trained with the proposed approaches.
- Re (B): The authors only use very rudimentary data augmentation techniques- mixup and contrast- and only compare to simple and internal baselines, even though there is a rich literature of recent data augmentation approaches that perform much better (e.g. even on the leaderboards recently: https://paperswithcode.com/sota/image-classification-on-cifar-100).
- Again to be clear: this reviewer is *not* saying that all papers need to compare with SOTA.  But if the goal is not to show strong real world performance / applicability as in (B), then the experiments section should at least accomplish something about elucidating core theoretical tradeoffs, insights, etc. as in (A).

---

> ### Author Response · Authors · 2020-11-24
> **Response to AnonReviewer1**
>
> We thanks for your constructive feedback and useful comments. Below are our responses to the comments.
>
> Re(A). There is no detailed experimental analysis beyond comparing end performance of models trained with the proposed approaches.
>
> A: We provide the performance in the experiments since it is the most important and interesting result as many deep learning literature did. Besides, we have an experiment of Mixup with 3 or 10 images, showing that when the bias level is high (strong augmentation), the proposed method still can outperform the standard Mixup. We will add more results for ablation study in the final version.
>
> Re(B). a rich literature of recent data augmentation approaches that perform much better.
>
> A: We agree with the reviewer's concern about additional experiments. However, the main goal of this paper focuses on the theoretical analysis of how to better utilize data augmentation, and we use the empirical evaluations to verify the theoretical findings. Besides, the comparison between heuristic algorithms and theoretical algorithms sometimes is not fair. For example, as one popular augmentation technique, Auto-Augment focuses on learning a good policy among augmentations while we consider how to better use a given augmentation. Besides, the training process of Auto-Augment usually needs much more running time than ours. Based on its original paper, Auto-Augment needs 5000 GPU hours to train Wide-ResNet-40-2 on CIFAR-10 for 120 epochs, while ours is much faster. But, we will add some comparisons between different heuristic augmentation techniques and the proposed methods in the final version.

---

### Official Review · AnonReviewer3 · 2020-10-28
**The authors propose a method to better utilize data augmentations especially when the bias between the original and augmented data distribution is large.**

**Rating:** 5
**Confidence:** 3

**Review:**

This paper proposes a method to improve the generalization of deep networks when the input is applied strong augmentations, i.e. mixup, resulting in large data bias. The authors proposes theoretical foundation behind their two methods, AugDrop and MixLoss and show their effectiveness on CIFAR10/CIFAR100 datasets. Although the paper has technical novelty and it only provides incremental gains on relatively small-size dataset. Below, you can find some of my comments/questions about the paper.

1. The paper has some typos and grammar mistakes. Some of them are "By contrast, we do not need original data is unlimited.", "it is sufficiently to optimize Lc(w)"

2. I feel like the notations used in the introduction is repeated in the beginning of the section 3. There is no need to repeat the same explanations and notations in section 3.

3. I am not sure about the definition of data bias mentioned in the paper. The bias is defined as the difference between the label of the augmented instance and original instance. In reality, there might be strong bias in the label-preserving augmentations, i.e. Jigsaw. I would like to get a clarification on this.

4. The paper only shows experiments only on CIFAR10 and CIFAR100. Potential positive results on ImageNet would have made the paper much convincing and stronger. This is because, in the past, many augmentation methods, i.e. CutOut, has been shown to perform well on CIFAR10/100 but not on ImageNet.

5. The improvements on CIFAR10/CIFAR100 are very marginal. There exists better data augmentation techniques, i.e. auto-augment, yielding better results on CIFAR10/100. It would be nice to include the results from different augmentation techniques into the experiments sections.

6. Similar to my point at 3, do the authors think that their method would work on an another augmentation method, i.e. jigsaw, that preserves the label? Or is it only constrained for the ones that changes the label.

7. What are the disadvantages of the proposed methods? What kind of complexity do they add, i.e. increased training complexity?

8. I understand that the authors try to show that in the case of very strong augmentations (mixup with 10 images), they can improve the baseline mixup augmentation. However, their performance on the mixup with 10 images is worse than the mixup with 3 images. Then, what exactly is the point of using very strong augmentations if we can not benefit from them?

---

> ### Author Response · Authors · 2020-11-24
> **Response to AnonReviewer3**
>
> We thank the reviewer for a detailed review and constructive feedback. We address the questions pointed out by the reviewer below:
>
> Q1. some typos and grammar mistakes.
>
> A: We have updated them in the revision.
>
> Q2. Notation.
>
> A: We have removed the notations in Section 3 of the revision.
>
> Q3. how about bias in the label-preserving augmentations?
>
> A:  Yes, we have this case in the paper. Due to the space limitation, we included the results of label-preserving augmentation in Appendix  A.
>
> Q4. experiments on ImageNet
>
> A:  We are conducting new experiments on ImageNet. We are going to add the experimental results to the final version.
>
> Q5. experiments of Auto-Augment
>
> A:  We would like to mention that the comparison between Auto-Augment and ours is not fair since (1) their algorithm is heuristic without any theoretical guarantee while ours focus on theoretical guarantees; (2) their algorithm focuses on learning a good policy among augmentations while we consider how to better use a given augmentation. Besides, Auto-Augment usually needs much more running time than ours for the entire training process. According to its original paper, Auto-Augment needs 5000 GPU hours to train Wide-ResNet-40-2 on CIFAR-10 for 120 epochs, while the proposed method is much faster. However, we can add such results from different heuristic augmentation techniques to the experiments in the final version.
>
> Q6. label-preserving augmentations
>
> A: Yes, we have the results of label-preserving augmentation in Appendix  A.
>
> Q7. Disadvantage of the proposed methods and the complexity.
>
> A: We found no obvious disadvantage. In theory, our methods do not increase training complexity.
>
> Q8. Mixup with 10 images.
>
> A: First, the results show the robustness of WeMix. Second, we want to examine how the level of bias (strong augmentation has large bias) influences the performances of the proposed methods.

---

### Official Review · AnonReviewer4 · 2020-10-28
**WeMix help train with augmented data**

**Rating:** 7
**Confidence:** 4

**Review:**

This paper illustrated two algorithms that correct bias in data augmentation: AugDrop and MixLoss. The authors illustrate their algorithms from a theoretical perspective and have mathematical proof the correctness of their algorithms.

The paper tackled the common problem on how to train with augmented data. it also half-opened a black-box of the training schemes. Authors' initial motivation can help the field significantly.

The concern I have is that if their method could be applied to a large range of applications since their experiment only based on some typical benchmarks, which already shows a good result in this field.

But generally, this paper is interesting and acceptable.

---

> ### Author Response · Authors · 2020-11-24
> **Response to AnonReviewer4**
>
> We would like to thanks for your review and your positive rate.
>
> About the experiments, we plan to add more results in other applications to the final version.

---

### Official Review · AnonReviewer2 · 2020-10-29
**A theoretical paper with a few implementation suggestions**

**Rating:** 4
**Confidence:** 2

**Review:**

##########################################################################

Summary:


In this work, the authors first prove a deep model can benefit from augmented data when the data bias is small. Then they propose two methods, namely "AugDrop" that corrects  and "MixLoss", that correct data bias by "constrained optimisation" and "modified loss function" respectively. Finally they show that these two methods can be combined and further improve the performance.

##########################################################################

Reasons for score:


I'm not an expert in the area of data augmentation, and I didn't check all the proofs and derivations in this paper. If I understand it correctly, the realisation of "AugDrop" is to omit data augmentation and train the model on original data, and the realisation of "MixLoss" is to train the model on both augmented and original data. The combined method, namely, "WeMix" is to run "MixLoss" first and then run "AugDrop". Overall, this paper is hard for me to follow. Yet another concern is that, the method was tested on CIFAR-10 and CIFAR-100, with "mixup" as the only competitor.

##########################################################################

Pros:

1. The paper tried to answer an important question "when will the deep models benefit from data augmentation?" They proved (though I didn't check the proof) that "a deep model can benefit from augmented data when the data bias is small". Based on this, they developed two methods "AugDrop" and "MixLoss".

2. The proposed method seems to be very easy to implement.

##########################################################################

Cons:


1. I'm not convinced by the current experiments: (i) it runs on CIFAR-10 and CIFAR-100 only (ii) the only competitor is "mix-up".

2. It wasn't quite clear (at least for me) why "AugDrop" and "MixLoss" can correct the data bias. Actually I have been lost at how the authors quantify the data bias.

##########################################################################

---

> ### Author Response · Authors · 2020-11-24
> **Response to AnonReviewer2**
>
> We thank the reviewer for reviewing our paper.
>
> Q1. about the experiments.
>
> We conducted experiments with two different types of augmentation. For label-preserving augmentation we used "contrast", while for label-mixing augmentation we used "mixup".  Since the main contribution of this paper is to explain how to better utilize data augmentation from a theoretical point of view, the main purpose of experimental evaluation is to verify the theoretical findings.  However, we expect to include more experiments in the final version.
>
>
> Q2. why "AugDrop" and "MixLoss" can correct the data bias?
>
> As we shown in the paper, AugDrop can correct the data bias by solving a constrained optimization problem so that it will enjoy small condition number to correct the data bias in the convergence analysis. MixLoss can correct data bias by using a modified loss $\mathcal{L}_c$ in (16). Specifically, the new augmented loss $\mathcal{L}_a$ can correct data bias since it has the same minimizer as original loss $\mathcal{L}$. By contrast, the minimizer of standard augmented loss $\mathcal{\widetilde L}$ can be very different from the minimizer of original loss $\mathcal{L}$.

---

### Decision · Program_Chairs · 2021-01-07
**Final Decision**

**Decision:**

Reject

**Comment:**

This work presents a new theoretically motivated data augmentation technique. Reviewers agreed that the theory was interesting and has value, however raised concerns regarding the experimental evaluation which was limited to the Cifar datasets. There was some discussion over whether or not a comparison with AutoAugment would be fair, the proposed method is theoretically motivated whereas AutoAugment takes significant compute to train. I agree with the authors that if the method doesn't outperform AutoAugment on CIFAR, this would not necessarily invalidate their results. Nonetheless the work would be significantly strengthened if it included results. on additional datasets to stress test the theory. I recommend the authors add additional supporting evidence and resubmit.